# DeepReShape: Redesigning Neural Networks for Efficient Private Inference

**Nandan Kumar Jha**                                    *nj2049@nyu.edu*
*New York University*

**Brandon Reagen**                                       *bjr5@nyu.edu*
*New York University*

**Reviewed on OpenReview:** *https://openreview.net/forum?id=iwCBWULItx*

## Abstract

Prior work on Private Inference (PI)—inferences performed directly on encrypted input—has focused on minimizing a network's ReLUs, which have been assumed to dominate PI latency rather than FLOPs. Recent work has shown that FLOPs for PI can no longer be ignored and incur high latency penalties. In this paper, we develop DeepReShape, a technique that optimizes neural network architectures under PI's constraints, optimizing for both ReLUs *and* FLOPs for the first time. The key insight is strategically allocating channels to position the network's ReLUs in order of their criticality to network accuracy, simultaneously optimizes ReLU and FLOPs efficiency. DeepReShape automates network development with an efficient process, and we call generated networks HybReNets. We evaluate DeepReShape using standard PI benchmarks and demonstrate a 2.1% accuracy gain with a $5.2\times$ runtime improvement at iso-ReLU on CIFAR-100 and an $8.7\times$ runtime improvement at iso-accuracy on TinyImageNet. Furthermore, we investigate the significance of network selection in prior ReLU optimizations and shed light on the key network attributes for superior PI performance.

## 1 Introduction

**Motivation** The increasing trend of cloud-based machine learning inferences has raised significant privacy concerns, leading to the development of private inference (PI). PI allows clients to send encrypted inputs to a cloud service provider, which performs computations without decrypting the data, thereby enabling inferences without revealing the data. Despite its benefits, PI introduces substantial computational and storage overheads (Mishra et al., 2020; Rathee et al., 2020) due to the use of complex cryptographic primitives (Demmler et al., 2015; Mohassel & Rindal, 2018; Patra et al., 2021).

Current PI frameworks attempt to mitigate these overheads by adopting hybrid cryptographic protocols and using additive secret sharing for linear layers (Mishra et al., 2020). This approach offloads homomorphic encryption tasks to an input-independent offline phase, achieving near plaintext speed for linear layers during the online phase. However, it fails to address the overheads of nonlinear functions (e.g., ReLU), which remain orders of magnitude slower than linear operations (Ghodsi et al., 2020).

In PI, Garbled Circuits (GCs)—a key cryptographic primitive that allows two parties to jointly compute arbitrary Boolean functions without revealing their data (Yao, 1986)—are used for private computation of nonlinear functions. In GC, nonlinear functions are first decomposed into a binary circuit (AND and XOR gates), which are then encrypted into truth tables (i.e., Garbled tables) for bitwise processing of inputs (Ball et al., 2016; 2019). The key challenge in GC is the storage burden: a single ReLU operation in GC requires 18 KiB of storage (Mishra et al., 2020), and networks with millions of ReLUs (e.g., ResNet50) can demand approximately ~100 GiB of storage for a single inference (Rathee et al., 2020). Additionally, computing all

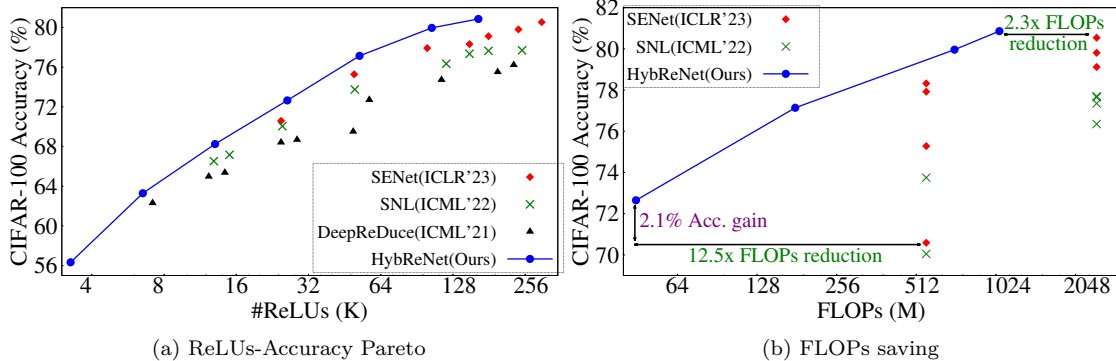

(a) ReLUs-Accuracy Pareto          (b) FLOPs saving

Figure 1: HybReNet outperforms state-of-the-art (SOTA) ReLU-optimization methods SENets(Kundu et al., 2023), SNL(Cho et al., 2022b), and DeepReDuce(Jha et al., 2021), achieving higher accuracy (CIFAR-100) and significant reduction in FLOPs while using fewer ReLUs (Table 6 illustrates the Pareto points specifics).

ReLUs in GC for a network like ResNet18 takes ∼21 minutes on TinyImageNet dataset (Garimella et al., 2023). Therefore, ReLUs are considered a primary source of storage and latency overheads in PI.

Prior work on PI-specific network optimization primarily focused on reducing nonlinear computation overheads, assuming linear operations (FLOPs) are effectively free. For instance, CryptoNAS (Ghodsi et al., 2020) and Sphynx (Cho et al., 2022a) employed neural architecture search for designing ReLU-efficient baseline networks without considering FLOPs implications. Similarly, PI-specific ReLU pruning methods (Cho et al., 2022b; Jha et al., 2021) made overly optimistic assumptions that all FLOPs can be processed offline without affecting real-time performance. The existing state-of-the-art (SOTA) in PI (Kundu et al., 2023) claimed that FLOPs cost is 343× less significant than ReLU cost. However, recent research (Garimella et al., 2023) has challenged these assumptions, demonstrating that FLOPs introduce significant latency penalties in end-to-end system-level PI performance[1].

Consequently, there is an emerging need to develop network design principles and optimization techniques that address both ReLU and FLOPs constraints in PI. This raises two critical questions: Can we leverage existing FLOPs reduction techniques and integrate them with PI-specific ReLU pruning methods? Second, how effective is it to employ PI-specific ReLU pruning techniques on FLOPs efficient networks, such as MobileNets (Howard et al., 2017; Sandler et al., 2018)?

**Challenges** Balancing ReLU efficiency with FLOPs efficiency is crucial for PI-specific network design optimization methods. SENet++ (Kundu et al., 2023) integrate FLOPs reduction technique with their ReLU pruning method and achieves (up to) 4× FLOPs reduction; however, at the expense of ReLU efficiency. The impact of existing FLOPs reduction methods on ReLU efficiency has not been extensively explored, and Jha et al. (2021) showed that FLOPs pruning methods tend to result in lower ReLU efficiency.

Furthermore, employing ReLU pruning on FLOPs-optimized networks results in inferior ReLU efficiency. For example, when ReLU pruning (Jha et al., 2021) employed on MobileNets, their ReLU efficiency remains consistently lower compared to standard networks (e.g., ResNet18) used in PI (see Figure 5(a)). Similarly, SOTA FLOPs efficient networks such as RegNet (Radosavovic et al., 2020) and ConvNeXt-V2 (Woo et al., 2023) exhibit suboptimal ReLU efficiency compared to the PI-tailored networks (see Figure 11).

This conflict between ReLU and FLOPs efficiency arises from the distinct layer-specific distribution of ReLUs and FLOPs in the network and their impact on network accuracy. In conventional CNNs, ReLUs are concentrated in the early layers, while ReLUs critical for the network's accuracy reside in deeper layers (Jha et al., 2021). ReLU pruning often removes many ReLUs from these early layers (Cho et al., 2022b; Jha et al., 2021), while FLOPs pruning targets the deeper layers due to their higher channel counts (He et al., 2020).

---

[1]In real-world scenarios, there is invariably some degree of inference arrival, and even at very low arrival rates, processing FLOPs offline becomes impractical due to limited resources and insufficient time. Consequently, FLOPs start affecting real-time performance, becoming more pronounced for networks with higher FLOPs. The FLOPs penalties can only be disregarded when there is zero inference arrival rate or when a homomorphic accelerator offering more than 1000× speedup is employed.

Moreover, designing ReLU efficient networks requires different network hyper-parameters than those needed for FLOPs efficient networks (See Table 15).

Another significant challenge in designing PI-tailored networks is identifying critical network attributes for PI efficiency. The effectiveness of PI-specific ReLU optimization techniques largely depends on the choice of input networks, leading to significant performance disparities not solely ascribed to FLOPs or accuracy discrepancies (refer to §3.2). Prior work on ReLU optimization offers limited insight into their network selection — SENets (Kundu et al., 2023) and SNL (Cho et al., 2022b) used WideResNet-22x8 for higher ReLU counts and ResNet18 for low ReLU counts. This leaves a gap in understanding whether networks with specific characteristics can maintain superior performance across various ReLU counts or if targeted ReLU counts dictate the desired network attributes.

The limitations of the existing ReLU-optimization techniques also impede the advancement of PI. Coarse-grained ReLU optimizations (Jha et al., 2021) encounter scalability issues, as their computational complexity varies linearly with the number of stages in a network. While fine-grained ReLU optimization (Cho et al., 2022b; Kundu et al., 2023) shows potential, its effectiveness is confined to specific ReLU distributions and tends to underperform in networks with higher ReLU counts or altered ReLU distribution (refer to §3.3).

**Our techniques and insights** To simultaneously optimize both the ReLU and FLOPs efficiency, we begin by critically evaluating existing design principles and posing a fundamental question: What essential insights need to be integrated into the design framework for achieving FLOPs efficiency without compromising the ReLU efficiency? Our sensitivity analysis of different network stages on ReLU and FLOPs efficiency reveals two key observations:

1. Increasing the network's width while positioning the network's ReLU based on their criticality to network's accuracy allows FLOPs reduction without sacrificing ReLU efficiency (Figure 4).

2. Widening channels in each network stages has distinct effect on network's overall ReLU and FLOPs efficiency (Figure 3(e, f)).

These insights led us to propose ReLU-equalization, a novel design principle that redistributes ReLUs in a conventional network by their order of criticality for network's accuracy (Figure 8), inherently accounting for the distinct effect of network stages on ReLU and FLOPs efficiency.

Our investigation into key network attributes for PI efficiency indicates that specific characteristics are essential for superior performance at different ReLU counts. We discovered that wider networks improve PI performance at higher ReLU counts. Whereas, at lower ReLU counts, the proportion of least-critical ReLUs in the network is crucial, especially when ReLU pruning is employed. Leveraging this insight, we achieve a significant, up to **45×**, FLOPs reduction at lower ReLU counts.

Building on the these insights, we develop DeepReShape, a framework to redesign the classical networks, with an efficient process of computational complexity $\mathcal{O}(1)$, and synthesize PI-efficient networks HybReNet. Our approach results in a substantial FLOPs reduction with fewer ReLUs, outperforming the SOTA in PI (Kundu et al., 2023). Precisely, we achieve a 2.3× ReLU and 3.4× FLOPs reduction at iso-accuracy, and a 2.1% accuracy gain with a **12.5×** FLOPs reduction at iso-ReLU on CIFAR-100 (see Figure 1). On TinyImageNet, we achieve **12.4×** FLOPs reduction at iso-accuracy compared to SOTA (see Table 7).

**Contributions** Our key contributions are summarized as follows.

1. Extensive characterization to identify the key network attributes for PI efficiency and demonstrate their applicability across a wide range of ReLU counts.
2. A novel design principle *ReLU-equalization*, and design of the *HybReNet* family of networks tailored to PI constraints. Moreover, we devise *ReLU-reuse*, a channel-wise ReLU dropping technique to systematically reduce the ReLU count by **16×**, allowing efficient ReLU optimization even at very low ReLU counts.
3. Rigorous evaluation of our proposed techniques against SOTA PI methods (Kundu et al., 2023; Cho et al., 2022b) and SOTA FLOPs efficient models (Woo et al., 2023; Radosavovic et al., 2020).

**Scope of the paper** This paper addresses the challenges of strategically dropping ReLUs from the convolutional neural networks (CNNs) without resorting to any approximated computations for nonlinearity. We exclude the models with complex nonlinearities, such as transformer-based models and FLOPs efficient

models like EfficientNet and MobileNetV3 [2], often relying on approximated nonlinear computations in PI. Also, we exclude the CrypTen-based PI in CNNs (Tan et al., 2021; Peng et al., 2023), as it operates under different security assumptions [3].

**Organization of the paper** Section 2 provides the relevant background on PI protocols, threat models, and network architecture, along with an overview of channel scaling methods and a categorization of PI-specific ReLU pruning methods. Section 3 comprehensively evaluates baseline network design and ReLU optimization strategies within the context of PI, outlining their limitations and our key observations. Section 4 introduces the DeepReShape method, followed by Section 5, which presents our experimental findings, and Section 6, summarizing the related work. Finally, we discuss the broader impact, limitations and future work Section 7.

## 2 Preliminary

**Private inference protocols and threat model** We use Delphi (Mishra et al., 2020) two-party protocols, as used in Jha et al. (2021); Cho et al. (2022b), for private inference. In particular, for linear layers, Delphi performs compute-heavy homomorphic operations (Gentry et al., 2009; Fan & Vercauteren, 2012; Brakerski et al., 2014; Cheon et al., 2017) in the offline phase (preprocessing) and additive secret sharing (Shamir, 1979) in the online phase, once the client's input is available. Whereas, for nonlinear (ReLU) layers, it uses garbled circuits (Yao, 1986; Ball et al., 2019). Further, similar to Liu et al. (2017); Juvekar et al. (2018); Mishra et al. (2020); Rathee et al. (2020), we assume an honest-but-curious adversary where parties follow the protocols and learn nothing beyond their output shares.

Note that the different sets of protocols for PI significantly affect the cost dynamics (communication, storage, and latency) for linear and nonlinear layers, thereby influencing network optimization goals. For instance, CoPriv (Zeng et al., 2023b) uses oblivious transfer (OT) for nonlinear operations and primarily optimizes convolution operations (i.e., FLOPs). Unlike OT, GCs offer constant round complexity and typically distribute more computational load to the server for garbling the circuit, reducing the client's computational burden (Demmler et al., 2015; Patra et al., 2021). In this work, we compare against prior approaches that use GCs for ReLUs, similar to the cryptographic setup of Delphi (Mishra et al., 2020).

**Architectural building blocks** Figure 2 illustrates a schematic view of a standard four-stage network with design hyperparameters. Similar to ResNet (He et al., 2016), it has a stem cell (to increase the channel count from 3 to $m$), followed by the network's main body (composed of linear and nonlinear layers, performing most of the computation), followed by a head (a fully connected layer) yielding the scores for the output classes. The network's main body is composed of a sequence of four stages, and the spatial dimensions of feature maps ($d_k \times d_k$) are progressively reduced by $2\times$ in each stage (except Stage1), and feature dimensions remain constant within a stage. We keep the structure of the stem cell and head fixed and change the structure of the network's body using design hyperparameters.

**Notations and definitions** Each stage is composed of identical blocks[4] repeated $\phi_1$, $\phi_2$, $\phi_3$, and $\phi_4$ times in Stage1, Stage2, Stage3, and Stage4 (respectively), and known as *stage compute ratios*. The output channels in stem cell ($m$) are known as *base channels*, and the number of channels progressively increases by a factor of $\alpha$, $\beta$, and $\gamma$ in Stage2, Stage3, and Stage4 (respectively), and we termed it as *stagewise channel multiplication factors*. The spatial size of the kernel is denoted as $f \times f$ (e.g., $3 \times 3$). These width and depth hyperparameters primarily determine the distribution of ReLUs and FLOPs in the network.

---

[2] Private inference on transformer-based models entail fundamentally different challenges (Chen et al., 2022b; Hao et al., 2022; Akimoto et al., 2023; Zheng et al., 2023; Hou et al., 2023; Gupta et al., 2023). CNNs predominantly employ crypto-friendly nonlinearities, e.g., ReLUs (and MaxPool, if at all used); while, transformers utilize complex nonlinearities like Softmax, GeLU, and LayerNorm. ReLUs in PI are precisely computed using Garbled-circuit (Mishra et al., 2020), whereas transformers often resort to approximations for their nonlinear computations due to performance objectives and numerical stability (Wang et al., 2022; Li et al., 2023; Zeng et al., 2023a; Zhang et al., 2023). Likewise, models such as EfficientNets (Tan & Le, 2019; 2021) and MobileNetV3 (Howard et al., 2019) incorporate Swish and Sigmoid nonlinearities to augment network expressiveness. These nonlinearities are approximated as discreet piecewise polynomials (Fan et al., 2022).

[3] CrypTen resembles a three-party framework since it adopts a Trusted Third Party (TTP) to produce beaver triples during the offline phase Knott et al. (2021). Consequently, the actual FLOPs overheads do not appear in end-to-end PI latency.

[4] Except the first block (in all but Stage1) which performs downsampling of feature maps by $2\times$.

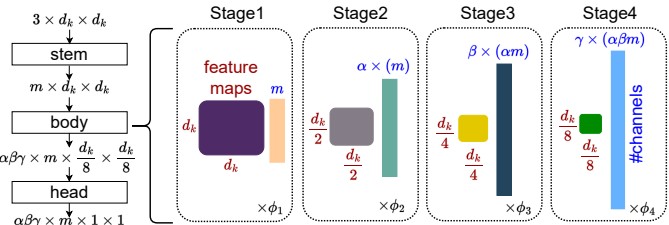

Figure 2: Depiction of architectural hyperparameters and feature dimensions in a four stage network. For ResNet18 $m = 64$, $\phi_1 = \phi_2 = \phi_3 = \phi_4 = 2$, and $\alpha = \beta = \gamma = 2$.

|  | Stage1 | Stage2 | Stage3 | Stage4 |
|---|---|---|---|---|
| $\frac{\#Params}{\#ReLU}$ | $m(\frac{f^2}{d_k^2})$ | $\alpha m(\frac{4f^2}{d_k^2})$ | $\alpha\beta m(\frac{16f^2}{d_k^2})$ | $\alpha\beta\gamma m(\frac{64f^2}{d_k^2})$ |
| $\frac{\#FLOPs}{\#ReLU}$ | $mf^2$ | $\alpha mf^2$ | $\alpha\beta mf^2$ | $\alpha\beta\gamma mf^2$ |

Table 1: Network's complexity (FLOPs and Params) per unit of nonlinearity varies with network's width, and independent of the network's depth. Consequently, *Wider network need fewer ReLUs for a given complexity,* compared to their deeper counterparts.

**Channel scaling methods** Broadly, channel scaling methods can be categorized into three categories (see Table 2). First, *Uniform channel scaling*, where $\alpha$, $\beta$, and $\gamma$ are set to 2 and channels are scaled either by scaling base channel counts (e.g., $m$=64 to $m$=128 in ResNets) or by a constant multiplication factor in all the network stages (e.g., $k$=10 in WideResNet22x10). We refer to their network variants as **BaseCh**, often used for FLOPs efficiency. Second, *homogeneous channel scaling*, where $\alpha$, $\beta$, and $\gamma$ are set identical, and channels in successive stages of the network are scaled by homogeneously augmenting these factors. For instance, $\alpha$, $\beta$, and $\gamma$ are set to 4 in CryptoNAS (Ghodsi et al., 2020) and Sphynx (Cho et al., 2022a)) for designing ReLU efficient baseline networks. We termed their network variants as **StageCh**. Third, *heterogeneous channel scaling*, where $\alpha$, $\beta$, and $\gamma$ are non-identical, and provides greater flexibility for balancing FLOPs and ReLU efficiency by scaling the channels in successive stages of the network differently.

**Criticality of ReLUs in a network** We employ the criticality metric $C_k$ from Jha et al. (2021) to quantify the significance of ReLUs' within a network stage for overall accuracy. Higher $C_k$ values indicate more critical ReLUs, while the least significant ReLUs are assigned a value of zero (see Table 10 and 11). Empirically, for a four-stage network like ResNet18 and its variants (BaseCh and StageCh), the ReLUs in Stage1 contribute the least and are the least critical, while those in Stage3 are the most critical for the network's accuracy.

Table 2: Comparison of channel scaling methods: Uniform channel scaling is a special case of homogeneous channel scaling where all stagewise channel multiplication factors ($\alpha$=$\beta$=$\gamma$) is identical and set to 2, and channels in network's stages are scaled by a constant factor (e.g., $k$=10 in in WideResNet22x10). In contrast, heterogeneous channel scaling differs by having non-identical factors, offering greater flexibility for balancing FLOPs and ReLU efficiency and meet the PI constraints.

| Channel Scaling Methods | Uniform | Homogeneous | Heterogeneous |
|---|---|---|---|
| Width Hyper-parameters | $\alpha = \beta = \gamma = 2$ | $\alpha = \beta = \gamma$ | $\neg(\alpha = \beta = \gamma)$ |
| Network Variants Naming | BaseCh | StageCh | HybReNet(**Proposed**) |
| Example Networks | WideResNet | CryptoNAS | HybReNets |
| Stage1 | $\begin{bmatrix} 3\times3,\ m\times k \\ 3\times3,\ m\times k \end{bmatrix} \times \phi_1$ | $\begin{bmatrix} 3\times3,\ m \\ 3\times3,\ m \end{bmatrix} \times \phi_1$ | $\begin{bmatrix} 3\times3,\ m \\ 3\times3,\ m \end{bmatrix} \times \phi_1$ |
| Stage2 | $\begin{bmatrix} 3\times3,\ 2m\times k \\ 3\times3,\ 2m\times k \end{bmatrix} \times \phi_2$ | $\begin{bmatrix} 3\times3,\ 4m \\ 3\times3,\ 4m \end{bmatrix} \times \phi_2$ | $\begin{bmatrix} 3\times3,\ \alpha m \\ 3\times3,\ \alpha m \end{bmatrix} \times \phi_2$ |
| Stage3 | $\begin{bmatrix} 3\times3,\ 4m\times k \\ 3\times3,\ 4m\times k \end{bmatrix} \times \phi_3$ | $\begin{bmatrix} 3\times3,\ 16m \\ 3\times3,\ 16m \end{bmatrix} \times \phi_3$ | $\begin{bmatrix} 3\times3,\ \beta(\alpha m) \\ 3\times3,\ \beta(\alpha m) \end{bmatrix} \times \phi_3$ |
| Stage4 |  |  | $\begin{bmatrix} 3\times3,\ \gamma(\alpha\beta m) \\ 3\times3,\ \gamma(\alpha\beta m) \end{bmatrix} \times \phi_4$ |

**Coarse-grained vs fine-grained ReLU optimization** The coarse-grained ReLU optimization method (Jha et al., 2021) removes ReLUs at the level of an entire stage or a layer in the network. Whereas fine-grained ReLU optimizations (Cho et al., 2022b; Kundu et al., 2023) target individual channels or activation. These approaches differ in performance, scalability, and configurability for achieving a specific ReLU count. The latter allows achieving any desired independent ReLU count automatically, while the former requires manual

adjustments based on the network's overall ReLU count and distribution. Nonetheless, the coarse-grained method demonstrates flexibility and adapting to various network configurations. In contrast, the fine-grained method exhibits less efficient adaptation and can lead to suboptimal performance (see §3.3).

## 3 Network Design and Optimization for Efficient Private Inference

In this section, we critically evaluate the current practices in baseline network design for efficient PI (§3.1), examines the selection of input networks for various ReLU-pruning methods (§3.2), and highlights the limitations of fine-grained ReLU optimization methods (§3.3). We further present our key observations, underscoring the significance of network architecture and ReLUs' distribution for end-to-end PI performance and motivate the need for redesigning the classical networks for efficient PI.

### 3.1 Addressing Pitfalls of Baseline Network Design for Efficient Private Inference

We begin by evaluating uniform and homogeneous channel scaling methods and their effectiveness in designing baseline networks for efficient PI. Subsequently, we investigate the impact of various channel scaling methods on the ReLUs' distribution within a network and motivate the need for heterogeneous channel scaling for optimizing FLOPs and ReLU counts simultaneously.

**The conventional uniform channel scaling leads to suboptimal ReLU efficiency** Table 1 shows that the (stagewise) complexity of the network, quantified as #FLOPs and #Params (Radosavovic et al., 2019), per units of ReLU nonlinearity scales linearly with base channel count $m$, while $\alpha$, $\beta$, and $\gamma$ introduce multiplicative effect. This implies that for a given network complexity, a network widened by augmenting $\alpha$, $\beta$, and $\gamma$ requires fewer ReLUs than the one widened by augmenting $m$. The uniform channel scaling in BaseCh networks, including WideResNet, often resorts to conservative $(\alpha, \beta, \gamma) = (2, 2, 2)$, which limits the potential ReLU efficiency benefit from wider networks.

**Homogeneous channel scaling offers superior ReLU efficiency until accuracy plateaus** In contrast to BaseCh networks, homogeneous channel scaling in StageCh networks significantly improves ReLU efficiency by removing the constraint on $(\alpha, \beta, \gamma)$ (Figure 3(a)). Nonetheless, the superiority of StageCh networks remains evident until reaching accuracy saturation, which varies with network configuration. In particular, as shown in Figure 3(b), accuracy saturation for StageCh networks of ResNet18, ResNet20, ResNet32, and ResNet56 models begins at $(\alpha, \beta, \gamma) = (4, 4, 4)$, $(5, 5, 5)$, $(5, 5, 5)$, and $(6, 6, 6)$, respectively, suggesting deeper StageCh network plateau at higher $(\alpha, \beta, \gamma)$ values. This observations challenge the assertion made in Ghodsi et al. (2020), that model capacity per ReLU peaks at $(\alpha, \beta, \gamma) = (4, 4, 4)$. Thus, determining the accuracy saturation point a priori is challenging, raising an open question: *To what extent can a network benefit from increased width for superior ReLU efficiency?* Moreover, can employing ReLU optimization on StageCh networks effectively address accuracy saturation?

**Homogeneous channel scaling alters the ReLUs' distribution distinctively than uniform scaling** We investigate the effect of uniform and homogeneous channel scaling on the ReLU distribution of networks. Unlike uniform scaling, which scales all layer ReLUs uniformly, homogeneous scaling leads to a distinct ReLU distribution, with deeper layers exhibiting more significant changes. As depicted in Figure 3 (c,d), there is a noticeable decrease in the proportion of Stage1 ReLUs, while Stage4 witnesses a significant increase. Given the ReLUs' criticality analysis in Table 10, this implies that the proportion of least-critical ReLUs is decreasing while the distribution of ReLUs among the other stages does not strictly adhere to their criticality order. This leads us to the following observation:

**Observation 1**: Homogeneous channel scaling reduces the percentage of least-critical ReLUs in the network.

**Heterogeneous channel scaling is required for optimizing ReLU and FLOPs efficiency simultaneously** To answer the question of potential benefits from wider networks, we perform a sensitivity analysis and evaluate the influence of each stagewise channel multiplication factor on the network's ReLU and FLOPs efficiency. We systematically vary one factor at a time, starting from 2, while other factors are held constant at 2, in ResNet18 with $m$=16. We observe that augmenting $\alpha$ and $\beta$ values improves ReLU efficiency; notably, the latter optimizes the performance marginally better than the former until a saturation point is

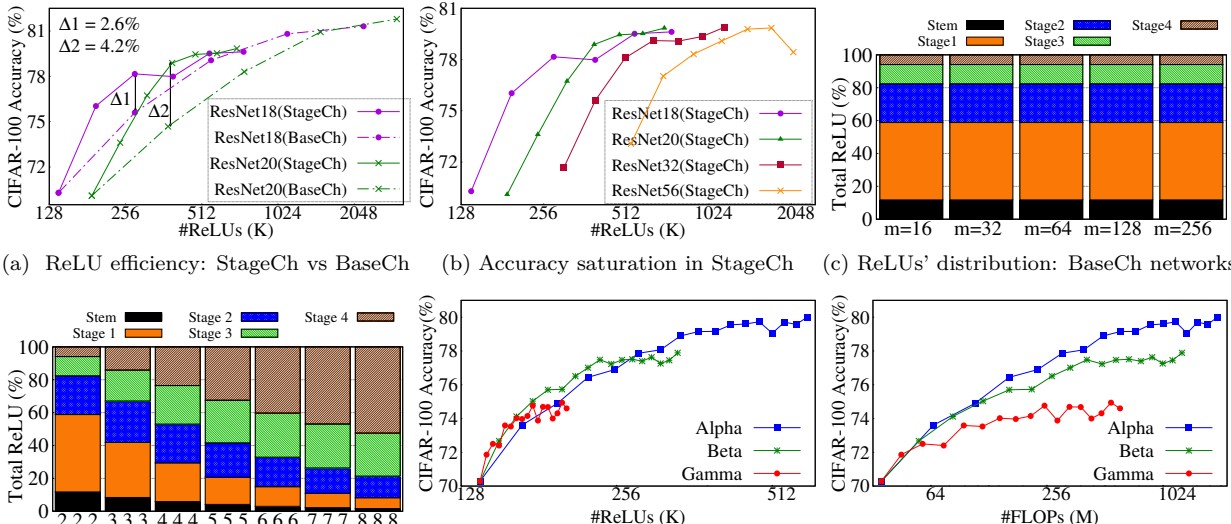

(a) ReLU efficiency: StageCh vs BaseCh  (b) Accuracy saturation in StageCh  (c) ReLUs' distribution: BaseCh networks

(d) ReLUs' distribution: StageCh networks (e) Sensitivity analysis: ReLU efficiency (f) Sensitivity analysis: FLOPs efficiency

Figure 3: (a) Homogeneous channel scaling in StageCh networks enables superior ReLU efficiency compared to uniform channel scaling in BaseCh networks; however, (b) the accuracy in StageCh networks tends to plateau unpredictably. (c,d) Unlike uniform channel scaling, homogeneous scaling reduces the proportion of least-critical ReLUs in StageCh networks. (e,f) Each network stage affects ReLU and FLOPs efficiency differently, requiring heterogeneous channel scaling for optimizing both ReLUs and FLOPs for efficient PI.

reached (see 3(c)). Whereas, FLOPs efficiency is most effectively improved by augmenting $\alpha$, outperforming $\beta$ enhancements while augmenting $\gamma$ values yields the worst FLOP efficiency (see 3(d)). This suggests that FLOPs in the deeper layers of StageCh networks can be regulated without impacting ReLU efficiency.

We note that the semi-automated designed networks RegNets (Radosavovic et al., 2020) employ heterogeneous channel scaling. However, they confine $1.5 \leq (\alpha, \beta, \gamma) \leq 3$ to optimize FLOPs efficiency, which in turn limits their ReLU efficiency (see Figure 11(c)). Thus, despite a line of seminal work on the network's width expansion (Zagoruyko & Komodakis, 2016; Radosavovic et al., 2019; Lee et al., 2019; Dollár et al., 2021), the approaches to leverage the potential benefits of increased width for simultaneously optimizing ReLUs and FLOPs efficiency remains an open challenge. The above analyses lead us to the following observation:

**Observation 2:** Each network stage *heterogeneously* impacts both ReLU and FLOPs efficiency, a nuanced aspect largely overlooked by prior channel scaling methods, rendering them inadequate for the simultaneous optimizing ReLUs and FLOPs counts for efficient private inference.

**Strategically scaling channels by arranging ReLUs in their criticality order can regulate the FLOPs in deeper layers without compromising ReLU efficiency** Following from the observations 1 and 2, we propose to scale the channels until all ReLUs are aligned in the criticality order. Thus, Stage3 dominates the distribution as it has the most critical ReLUs, followed by Stage2, Stage4, and Stage1 (Table 10). Unlike StageCh networks, widening beyond the point where the ReLUs are aligned in their criticality order does not alter their relative distribution (Figure 4(a)). This leads to higher $\alpha$ and $\beta$ values, which boost ReLU efficiency, with restrictive $\gamma$ ($\gamma < 4$) regulating FLOPs in deeper layers, promoting FLOP efficiency.

Consequently, our approach of heterogeneous channel scaling achieves ReLU efficiency on par with StageCh networks with fewer FLOPs. Figure 4(b,c) demonstrates that the ReLUs' criticality-aware ResNet18 network 5x5x3x maintains similar ReLU efficiency with a 2× reduction in FLOPs compared to the StageCh network 5x5x5x. This FLOP reduction is consistently attained across the entire spectrum of ReLU counts, employing both fine-grained and coarse-grained ReLU optimization. These results lead to the following observation:

**Observation 3:** ReLUs' criticality-aware network widening method optimizes FLOPs efficiency without sacrificing the ReLU efficiency, which meets the demands of efficient PI.

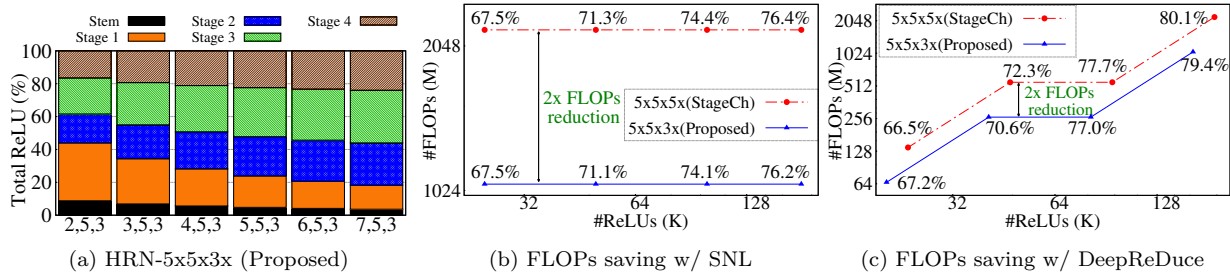

(a) HRN-5x5x3x (Proposed)  (b) FLOPs saving w/ SNL  (c) FLOPs saving w/ DeepReDuce

Figure 4: (a) Unlike StageCh networks, once the network's ReLUs are aligned in their criticality order, here at point $(\alpha, \beta, \gamma)=(5, 5, 3)$, increasing $\alpha$ does not alter their relative distribution. (b,c) ReLUs' criticality-aware network widening method saves $2\times$ FLOPs by regulating the FLOPs in deeper layers while maintaining ReLU efficiency over a wide range of ReLU counts.

### 3.2 Addressing Fallacies in Network Selection for ReLU Optimization

In this section, we explore the crucial aspects of selecting appropriate input networks for various ReLU pruning methods and perform a detailed experimental analysis to identify network attributes crucial for PI efficiency across different ReLU counts. This study aims to bridge the knowledge gap for designing efficient baseline networks tailored to ReLU pruning methods.

**Selecting the appropriate input network for ReLU optimization methods is far from intuitive** Table 3 lists input networks used in previous ReLU optimization methods with their relevant characteristics, while Figure 5 demonstrates how different input networks affect the performance of coarse (DeepReDuce) and fine-grained (SNL) ReLU optimization methods. For the former, accuracy differences of **12.9%** and **11.6%** are observed at higher and lower iso-ReLU counts. *These differences cannot be ascribed to the FLOPs or accuracy of the baseline network alone.* For instance, ResNet18 outperforms WideResNet22x8 despite having $4.4\times$ fewer FLOPs and a lower baseline accuracy, and ResNet32 outperforms VGG16 even though the latter has $4.76\times$ more FLOPs and a higher baseline accuracy.

Likewise, fine-grained ReLU optimization (SNL) exhibits significant accuracy differences when employed on ResNets and WideResNets, especially at lower ReLU counts, as shown in Figure 5(b). While WideResNet models outperform beyond 200K ReLUs, there are 3.2% and 4.6% accuracy gaps at 25K and 15K ReLUs between ResNet18 and WideResNet16x8. The above empirical observation led to the following observation:

**Observation 4:** Performance of ReLU optimization methods, whether coarse or fine-grained, strongly correlates with the choice of input networks, leading to substantial performance disparities.

| ReLU optimization method | Input networks |
|---|---|
| Delphi (Mishra et al., 2020) | ResNet32 |
| SAFENets (Lou et al., 2021) | ResNet32, VGG16 |
| DeepReDuce (Jha et al., 2021) | ResNet18 |
| SNL (Cho et al., 2022b) | ResNet18, WRN22x8 |
| SENet (Kundu et al., 2023) | ResNet18, WRN22x8 |

| | ResNet32 | ResNet18 | WRN22x8 | VGG16 |
|---|---|---|---|---|
| FLOPs | 70M | 559M | 2461M | 333M |
| ReLUs | 303K | 557K | 1393K | 285K |
| Acc | 71.67% | 79.06% | 81.27% | 75.08% |

Table 3: Baseline networks used for advancing ReLU-Accuracy Pareto (CIFAR-100) in prior PI-specific ReLU optimization methods.

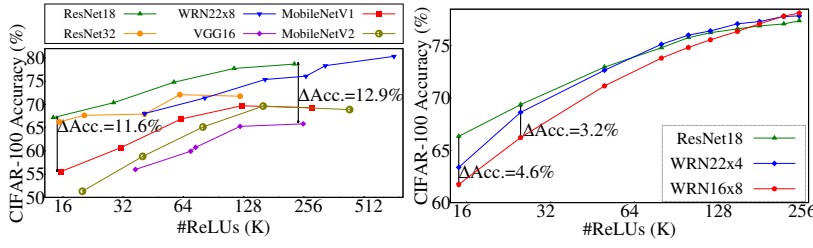

(a) DeepReDuce at iso-ReLU  (b) SNL at iso-ReLU

Figure 5: ReLU optimization, whether coarse or fine-grained, performance exhibits significant disparities based on the input networks.

**Key network attributes for PI efficiency vary across targeted ReLU counts** To identify the key network attributes for PI efficiency across a wide range of ReLU counts, we examine three ResNet18 variants with identical ReLU counts but different ReLUs' distribution and FLOPs counts (Table 4). These are realized by channel reallocation, and the configurations 2x2x2x($m=32$), 4x4x4x($m=16$), and 3x7x2x($m=16$) correspond to stagewise channel counts as [32,64,128,256], [16, 64, 256, 1024], and [16, 48, 336, 672] respectively. We analyze their performance using the DeepReDuce and SNL ReLU optimization, as shown in Figure 6.

A consistent trend emerges from both ReLU optimization methods: Wider models 4x4x4x($m$=16) and 3x7x2x($m$=16) outperform 2x2x2x($m$=32) at higher ReLU counts; however, even with $\approx 4\times$ fewer FLOPs, 2x2x2x($m$=32) excel at lower ReLU counts. This superior performance stems from the higher percentage (58.82%) of least-critical (Stage1) ReLUs in 2x2x2x($m$=32). When targeting low ReLU counts, ReLU optimization methods primarily drop ReLUs from Stage1 (Jha et al., 2021; Cho et al., 2022b; Kundu et al., 2023). Thus, networks with a higher percentage of Stage1 ReLUs preserve more ReLUs from critical stages, mitigating accuracy degradation. Furthermore, this emphasizes the importance of strategically allocating channels, even when aiming for higher ReLU counts: 3x7x2x($m$=16) matches the ReLU efficiency of 4x4x4x($m$=16) with 30% fewer FLOPs by allocating more channels to Stage3 and fewer to Stage4.

| Model | Acc(%) | FLOPs | ReLUs | Stagewise ReLUs' distribution | | | |
|---|---|---|---|---|---|---|---|
| | | | | Stage1 | Stage2 | Stage3 | Stage4 |
| 2x2x2x(m=32) | 75.60 | 141M | 279K | 58.82% | 23.53% | 11.76% | 5.88% |
| 4x4x4x(m=16) | 78.16 | 661M | 279K | 29.41% | 23.53% | 23.53% | 23.53% |
| 3x7x2x(m=16) | 78.02 | 466M | 260K | 31.50% | 18.90% | 33.07% | 16.54% |

Table 4: A case study to investigate the Capacity-Criticality-Tradeoff: Three Iso-ReLU ResNet18 networks with different ReLUs' distribution and FLOPs count, achieved by reallocating channels per stage. The baseline accuracy is for CIFAR-100 dataset.

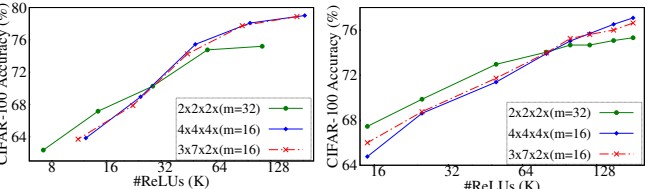

(a) DeepReDuce at iso-ReLU     (b) SNL at iso-ReLU

Figure 6: Capacity-Criticality-Tradeoff results: Figures (a) and (b) show the ReLU-Accuracy tradeoff for networks in Table 4 using DeepReDuce and SNL.

The above findings offer insight into the network selection for prior ReLU optimization methods. Specifically, the choice of WRN22x8 (with 48.2% Stage1 ReLUs) for higher ReLU counts while ResNet18 for lower ReLU counts in fine-grained ReLU optimization (Cho et al., 2022b; Kundu et al., 2023). Moreover, it also explains the accuracy trends depicted in Figure 5(b), the higher the Stage1 ReLU proportion (58.8% for ResNet18, 47.7% for WRN22x4, and 43.9% for WRN16x8), the higher the accuracy at lower ReLU counts.

Interestingly, we note that the above networks with a higher percentage of least-critical (Stage1) ReLUs inherently have fewer overall ReLUs (e.g., 1392.6K for WRN22x8 and 557K ResNet18). This might suggest that these networks utilize their ReLUs more effectively, especially when there are fewer ReLUs, leading them to excel at lower ReLU counts. However, a counter-example in Appendix E.2 reaffirms our conclusion for the key factor driving PI performance at lower ReLU counts. We further investigate the Capacity-Criticality-Tradeoff in Appendix E.1, and the additional results are shown in Figure 18. These analyses lead to the following observation:

**Observation 5:** Wider networks are superior only at higher ReLU counts, while networks with higher percentage of least-critical ReLUs outperform at lower ReLU counts (Capacity-Criticality-Tradeoff).

## 3.3 Mitigating the Limitations of Fine-grained ReLU Optimization

We now investigate the limitations of fine-grained ReLU optimization methods, often outperforming coarse-grained methods in conventional networks, and discuss the strategies to mitigate these limitations. This study aims to assess the efficacy of fine-grained methods beyond the conventional networks, especially with atypical ReLU distributions, for instance, when heterogeneous channel scaling is employed for simultaneously optimizing ReLU and FLOPs (see observation 3).

**Fine-grained ReLU optimization is not always the best choice** While fine-grained ReLU optimization has demonstrated its effectiveness in classical networks such as ResNet18 and WideResNet, especially when Stage1 dominates the network's ReLU distribution (Cho et al., 2022b; Kundu et al., 2023), its advantages are not universal. To better understand its range of efficacy, we compared it against DeepReDuce on PI-amenable wider models: 4x4x4x($m$=16) and 3x7x2x($m$=16) (Table 4).

As shown in Figure 7(a) and 7(b), DeepReDuce outperforms SNL by a significant margin (up to 3%-4%). This suggests that the benefits of fine-grained ReLU optimization are highly dependent on specific ReLU distributions, and it reduces when Stage1 does not dominate the network's ReLU distribution. This trend is also observed in ReLU criticality-aware networks, where Stage3 dominates the distribution of ReLUs (see

Figure 20). This empirical evidence collectively suggests that *fine-grained ReLU optimization might limit the benefits of increased network complexity* introduced through stagewise channel multiplication enhancements. Nonetheless, the performance gap is less pronounced when the network's overall ReLU count is reduced by half by using ReLU-Thinning (Jha et al., 2021), which drops the ReLUs from alternate layers.

| C100 | | Baseline | 220K | 180K | 150K | 120K | 100K | 80K | 50K |
|---|---|---|---|---|---|---|---|---|---|
| ResNet18 (557.06K) | Vanilla | 78.68 | 77.09 | 76.9 | 76.62 | 76.25 | 75.78 | 74.81 | 72.96 |
| | w/ Th. | 76.95 | 77.03 | 76.92 | 76.54 | 76.59 | 75.85 | 75.72 | 74.44 |
| | Δ | -1.73 | -0.06 | 0.02 | -0.08 | 0.34 | 0.07 | 0.91 | 1.48 |
| ResNet34 (966.66K) | Vanilla | 79.67 | 76.55 | 76.35 | 76.26 | 75.47 | 74.55 | 74.17 | 72.07 |
| | w/ Th. | 79.03 | 77.94 | 77.65 | 77.67 | 77.32 | 76.69 | 76.32 | 74.50 |
| | Δ | -0.64 | 1.39 | 1.30 | 1.41 | 1.85 | 2.14 | 2.15 | 2.43 |
| WRN22x8 (1392.64K) | Vanilla | 80.58 | 77.58 | 76.83 | 76.15 | 74.98 | 74.38 | 73.16 | 71.13 |
| | w/ Th. | 79.59 | 78.91 | 78.6 | 78.41 | 78.05 | 77.22 | 75.94 | 72.74 |
| | Δ | -0.99 | 1.33 | 1.77 | 2.26 | 3.07 | 2.84 | 2.78 | 1.61 |

Table 5: A significant accuracy boost (on CIFAR-100) is achieved when ReLU-Thinning is employed prior to SNL, despite the less accurate ReLU-Thinned models. Δ = Acc(w/ Th.)-Acc(Vanilla).

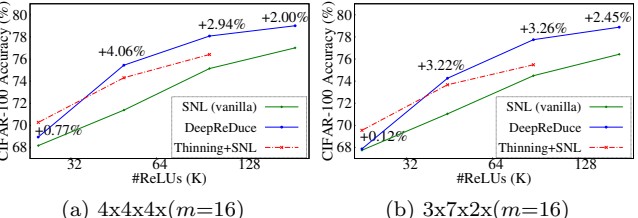

(a) 4x4x4x($m=16$)      (b) 3x7x2x($m=16$)

Figure 7: DeepReDuce outperforms SNL by a significant margin (**up to 4%**) when altering network's ReLUs distribution; however, using SNL on ReLU-Thinned networks reduces the accuracy gap.

**Narrowing the search space improves the performance of fine-grained ReLU optimization** To further examine the efficacy of ReLU-Thinning for classical networks, we adopt a *hybrid* ReLU optimization approach, and ReLU-Thinning is employed before SNL optimization. Surprisingly, *even when baseline Thinned models are less accurate*, a significant accuracy boost (up to **3%** at iso-ReLUs) is observed, which is more pronounced for networks with higher #ReLUs (ResNet34 and WRN22x8, in Table 5). Since ReLU-Thinning drops the ReLUs from the alternate layers, *irrespective of their criticality*, its integration into existing ReLU optimization methodologies would not impact their overall computational complexity and remains effective for reducing the search space to identify critical ReLUs. This leads us to the following observation:

**Observation 6:** While altering the network's ReLU distribution can lead to suboptimal performance in fine-grained ReLU optimization, ReLU-Thinning emerges as an effective solution to bridge the performance gap, also beneficial for classical networks with higher overall ReLU counts.

# 4   DeepReShape

Drawing inspiration from the above observations and insights, we propose a novel design principle termed *ReLU equalization* (Figure 8) and re-design classical networks. This led to the development of a family of models *HybReNet*, tailored to the needs of efficient PI (Table 16). Additionally, we propose *ReLU-reuse*, a (structured) channel-wise ReLU dropping method, enabling efficient PI at very low ReLU counts.

## 4.1   ReLU Equalization and Formation of HybReNet

Given a baseline input network, where ReLUs are not necessarily aligned in their criticality order, ReLU equalization redistributes the network's ReLUs in their criticality order, meaning the (most) least critical stage has a (highest) lowest fraction of the network's total ReLU count (Figure 8). Equalization is achieved by an iterative process, as outlined in Algorithm 1. In each iteration, the relative distribution of ReLUs in two stages is aligned in their criticality order by adjusting either their depth or width or both hyperparameters.

Specifically, for a network of $D$ stages and a predetermined criticality order, given compute ratios $\phi_1$, $\phi_2$, ..., $\phi_D$ and stagewise channel multiplication factors $\lambda_1$, $\lambda_2$, ..., $\lambda_{(D-1)}$, the ReLU equalization algorithm outputs a compound inequality after $D$-1 iterations. We now employ Algorithm 1 on a standard four-stage ResNet18 model with the given criticality order as (from highest to lowest): Stage3 > Stage2 > Stage4 > Stage1 (refer to Table 10). During the equalization process, only the model's width hyper-parameters are adjusted, as wider models tend to be more ReLU efficient. Consequently, the algorithm yields the following expression:

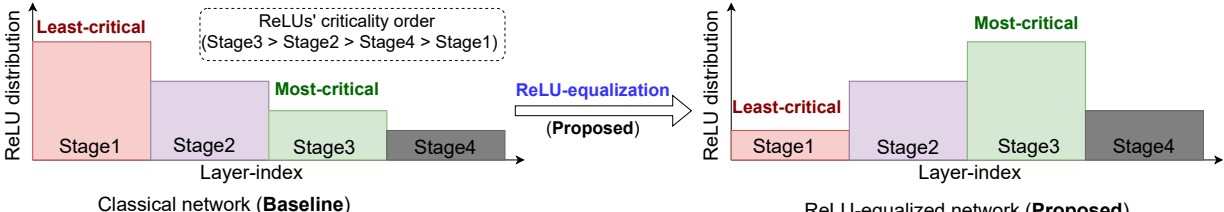

Figure 8: Illustration of ReLU-equalization: Unlike classical networks (e.g., ResNet), where ReLUs' are not positioned is their criticality order, ReLU-equalization aligns network's ReLUs in their criticality order.

---

**Algorithm 1** ReLU equalization

---

**Input:** Network $Net$ with stages $S_1,...,S_D$; $C$ a sorted list of most to least critical stage; stage-compute ratio $\phi_1,...,\phi_D$; and stagewise channel multiplication factors $\lambda_1,..., \lambda_{(D-1)}$.
**Output:** ReLU-equalized versions of network $Net$.

1: **for** $i = 1$ **to** $D$-1 **do**
2:     $S_k = C[i]$                                                                     ▷ $C[1]$ is most critical stage
3:     $S_t = C[i+1]$                                                                ▷ $C[2]$ is second-most critical stage
4:     **while** #ReLUs$(S_k) >$ #ReLUs$(S_t)$ **do**    ▷ ReLUs in two stages are aligned in their criticality order
5:         $\frac{\phi_k \times \left(\prod_{j=1}^{k-1} \lambda_j\right)}{2^{k-1}} > \frac{\phi_t \times \left(\prod_{j=1}^{t-1} \lambda_j\right)}{2^{t-1}}$    ▷ Rearranging ReLUs by adjusting width and depth parameters
6:     **end while**
7: **end for**
8: **return** A set of $\phi_1,...,\phi_D$ and $\lambda_1,...,\lambda_{(D-1)}$ that satisfies the compound inequality: #ReLUs$(C[1]) >$ #ReLUs$(C[2]) > ... >$ #ReLUs$(C[D-1]) >$ #ReLUs$(C[D])$

---

$$\#ReLUs(S_3) > \#ReLUs(S_2) > \#ReLUs(S_4) > \#ReLUs(S_1)$$

$$\implies \phi_3\left(\frac{\alpha\beta}{16}\right) > \phi_2\left(\frac{\alpha}{4}\right) > \phi_4\left(\frac{\alpha\beta\gamma}{64}\right) > \phi_1$$

ReLU equalization through width ($\phi_1 = \phi_2 = \phi_3 = \phi_4 = 2$, and $\alpha \geq 2, \beta \geq 2, \gamma \geq 2$) :

$$\implies \frac{\alpha\beta}{16} > \frac{\alpha}{4} > \frac{\alpha\beta\gamma}{64} > 1 \implies \alpha\beta > 16, \ \alpha > 4, \ \alpha\beta\gamma > 64, \ \beta > 4, \ \beta\gamma < 16, \ \text{and} \ \gamma < 4$$

Solving the above compound inequalities provides the following $(\beta,\gamma)$ pairs and the range of $\alpha$ :

The $(\beta,\gamma)$ pairs are: $(5,2) \ \& \ \alpha \geq 7$; $(5,3) \ \& \ \alpha \geq 5$; $(6,2) \ \& \ \alpha \geq 6$; $(7,2) \ \& \ \alpha \geq 5$

We obtain four pairs of $(\beta, \gamma)$, each having a range of $\alpha$ value. We choose the smallest $\alpha$ needed for ReLU equalization, as increasing $\alpha$ beyond this point does not improve the performance when ReLU optimization is used; also, the relative distribution of ReLUs remains stable (see Appendix A). Thus, we achieve four baseline HybReNets: HRN-5x5x3x, HRN-5x7x2x, HRN-6x6x2x, and HRN-7x5x2x. The architectural details of these four HRNs are presented in Table 16.

## 4.2 ReLU-reuse

We further refine the baseline network's architecture to increase ReLU nonlinearity utilization by introducing *ReLU-reuse*, which selectively applies ReLUs to a contiguous subset of channels while the remaining channels reuse them. *This approach differs from previous channel-wise ReLU optimizations*, where channels are either uniformly scaled down throughout the network (Jha et al., 2021) or only a subset of channels utilize ReLUs without reusing them (Cho et al., 2022b). Our ReLU-reuse mechanism allows for efficient PI at extremely low ReLU counts (e.g., 3.2K ReLUs on CIFAR-100 dataset).

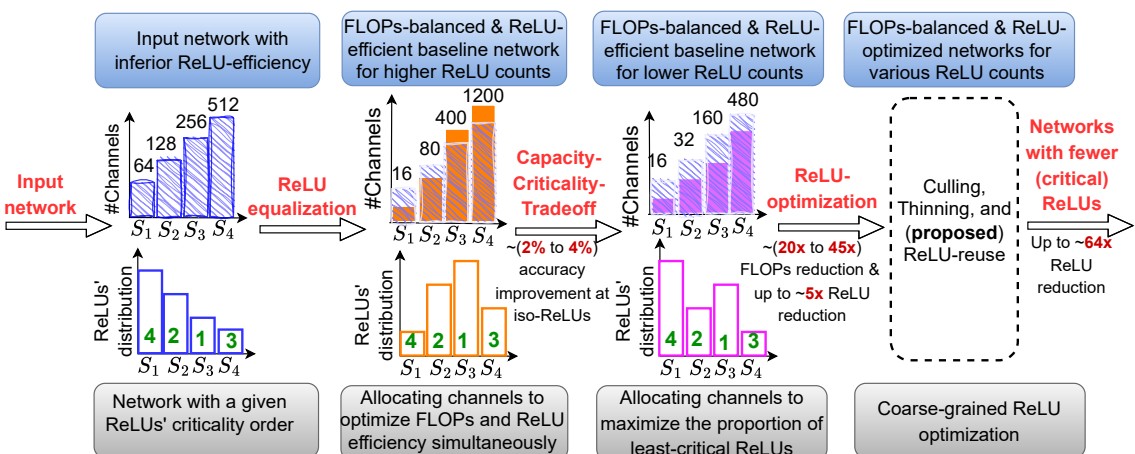

Figure 10: The DeepReShape network redesigning pipeline. ReLU's criticality-aware strategic allocation of channels (gray boxes) outputs FLOPs-balanced ReLU-efficient baseline networks for various ReLU counts (blue boxes). Numbers in green denote criticality order (Stage3 is most critical).

Specifically, feature maps of the layer are divided into $N$ groups, and ReLUs are employed only in the last group (Figure 9). However, increasing the value of $N$ results in a significant accuracy loss despite $1 \times 1$ convolution being employed for cross-channel interaction. This is likely due to the loss of cross-channel information arising from more divisions in the feature maps (see our ablation study in Table 9). To address this issue, we devise a mechanism that decouples the

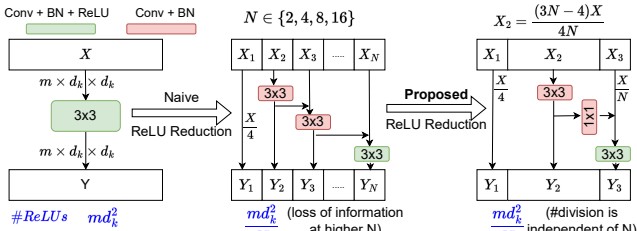

Figure 9: Proposed ReLU-reuse where ReLUs are *selectively* reused across channels, reducing #ReLUs up to $16\times$.

number of divisions in feature maps from the ReLU reduction factor $N$. Precisely, one-fourth of channels are utilized for feature reuse, while a $Nth$ fraction of feature maps are activated using ReLUs, and the remaining feature maps are processed solely with convolution operations, resulting in only three groups. It is important to note that using the ReLUs in the last group of feature maps *increases the effective receptive field* as these neurons can consider a larger subset of feature maps using the skip connections (Gao et al., 2019).

### 4.3 Putting it All Together

We developed the DeepReShape framework to re-design the classical networks for efficiency PI across a wide range of ReLU counts. Figure 10. Given an input network with a specific ReLUs' criticality order, the ReLU-equalization step aligns the network's ReLU in their criticality order by adjusting width hyper-parameters. This step allows for maximizing ReLU efficiency without incurring superfluous FLOPs by allocating fewer channels in the initial stages and increasing them in the deeper stages. In the second step, following the Criticality-Capacity-Tradeoff, the width is adjusted such that Stage1 dominates the ReLUs' distribution. This is achieved by a straightforward step: setting $\alpha=2$ in the ReLU-equalized networks since decreasing $\alpha$ results in an increased percentage of Stage1 ReLUs, and distribution of ReLUs in all but Stage1 follow their criticality order (see Table 11). This step allows for a substantial FLOP reduction, up to $45\times$, by allocating fewer channels in all the stages. We call the networks resulting from step1 and step2 as HybReNets (HRNs). The baseline HRNs from step2 are: HRN-2x5x3x, HRN-2x7x2x, HRN-2x6x2x, and HRN-2x5x2x (Table 17).

**ReLU-optimization steps for HybReNets** We choose to employ coarse-grained ReLU optimization steps in HRNs, as they outperform fine-grained ReLU optimization when the ReLU distribution undergoes changes in traditional networks, as shown in Figure 7 and Appendix F. In particular, we eliminate all the ReLUs from Stage1 (ReLU Culling) if it dominates the network's overall ReLU distribution, e.g., HRNs with $\alpha=2$. For subsequent stages, we utilize ReLU-Thinning, which removes ReLUs from alternate layers

without considering their criticality. We further reduce the ReLU count by implementing ReLU-reuse with an appropriate reduction factor (see Algorithm 2).

**Complexity analysis of HybReNet design** For a $D$ stage network with a predefined criticality order for stagewise ReLUs, the process of ReLU equalization typically involves considering $2D$-1 hyperparameters, including $D$ stage compute ratios and $D$-1 stagewise channel multiplication factors. However, for HRNs, this hyperparameter count is reduced to $D$-1 since ReLU equalization is achieved solely by modifying the network's width. Unlike SOTA network designing methods (Radosavovic et al., 2020; Liu et al., 2022), which build networks from scratch, the hyperparameters involved in ReLU equalization are determined by solving a compound inequality, eliminating the need for additional network training. That is, to narrow down the design search space provided by bounds on $\alpha$, $\beta$, and $\gamma$, we select the minimum values of these hyper-parameters that satisfy the ReLU equalization conditions. Thus, our method leverages the existing network designs and optimizes them under PI constraints rather than designing them from scratch. Consequently, the complexity of designing HRNs can be characterized as $\mathcal{O}(1)$. A detailed discussion is included in Appendix H.5.

Additionally, employing coarse-grained ReLU optimization does not exacerbate the complexity of HRNs. This is due to the positioning of ReLUs in HRNs based on their criticality order, which necessitates only a single iteration (see Algorithm 2). In contrast, when ReLUs in the input network are organized without regard to their criticality order (e.g., classical networks such as ResNets and WideResNets), a single iteration produces suboptimal results, requiring $D$-1 iterations (Jha et al., 2021). Thus, the complexity of ReLU optimization for HRNs is reduced to $\mathcal{O}(1)$ from $\mathcal{O}(D)$.

## 5 Experimental Results

**Analysis of HybReNets Pareto points** Figure 1 shows that HybReNet advances the ReLU-accuracy Pareto with a substantial reduction in FLOPs – a factor overlooked in prior PI-specific network optimization. We present a detailed analysis of network configurations and ReLU optimization steps and quantify their benefits for ReLUs and FLOP reduction. We use ResNet18-based HRN-5x5x3x for ReLU-accuracy comparison with SOTA PI methods in Figure 1, as its FLOPs efficiency is superior to other HRNs (Table 16).

Table 6: Network configurations and ReLU optimization steps used for the Pareto points in Figure 1. Accuracies (CIFAR-100) are separately shown for KD (Hinton et al., 2015) and DKD (Zhao et al., 2022), highlighting the benefits of improved architectural design and distillation method.(Re2 denotes ReLU-reuse)

| HybReNet | $m$ | ReLU optimization steps | | | #ReLU | #FLOPs | Accuracy(%) | | Acc./ReLU |
|----------|-----|--------|----------|-----|-------|--------|------|------|----------|
| | | Culled | Thinned | Re2 | | | KD | DKD | |
| 5x5x3x | 16 | NA | S1+S2+S3+S4 | NA | 163.3K | 1055.4M | 79.34 | 80.86 | 0.50 |
| 2x5x3x | 32 | S1 | S2+S3+S4 | NA | 104.4K | 714.1M | 77.63 | 79.96 | 0.77 |
| 2x5x3x | 16 | S1 | S2+S3+S4 | NA | 52.2K | 178.5M | 74.98 | 77.14 | 1.48 |
| 2x5x3x | 8 | S1 | S2+S3+S4 | NA | 26.1K | 44.6M | 70.36 | 72.65 | 2.78 |
| 2x5x3x | 16 | S1 | S2+S3+S4 | 4 | 13.1K | 121.6M | 67.30 | 68.25 | 5.23 |
| 2x5x3x | 16 | S1 | S2+S3+S4 | 8 | 6.5K | 130.5M | 62.68 | 63.29 | 9.70 |
| 2x5x3x | 16 | S1 | S2+S3+S4 | 16 | 3.2K | 137.2M | 56.24 | 56.33 | 17.26 |

The key takeaway from Table 6 is that tailoring the network features for PI constraint significantly reduces FLOPs and ReLUs. Specifically, lowering $\alpha$ value and base channel count led to **23.6×** fewer FLOPs in HRN-2x5x3x($m$=8), compared to HRN-5x5x3x($m$=16). Furthermore, we notice a significant accuracy boost by employing a simple yet efficient logit-based distillation method DKD (Zhao et al., 2022), as the ReLU-reduced models greatly benefit from decoupling the target and non-target class distillation.

**HybReNets outperform state-of-the-art in private inference** Table 7 presents competing design points for SENet (Kundu et al., 2023) and SNL (Cho et al., 2022b), and we select HybReNet points (see Table 6 and Table 13 for configuration and optimization details) offering both accuracy and latency benefits for a fair comparison. The runtime breakdown is presented as homomorphic (HE) latency (Brakerski et al., 2014),

Table 7: Comparison of HybReNet with SOTA in private inference: SENet (Kundu et al., 2023) and SNL (Cho et al., 2022b). HybReNet exhibits superior ReLU and FLOPs efficiency and achieve a substantial reduction in latency. #Re and #FL denote ReLU and FLOPs counts; Acc. is top-1 accuracy; Lat. is the runtime for one private inference, including Homomorphic (HE) and Garbled-circuit(GC) latencies.

| | | SOTA in Private Inference | | | | | | HybReNet(**Ours**) | | | | | | Improvements | | | | | |
|---|---|---|---|---|---|---|---|---|---|---|---|---|---|---|---|---|---|---|---|
| | | #Re | #FL | Acc. | HE | GC | Lat. | #Re | #FL | Acc. | HE | GC | Lat. | #Re | #FL | Acc. | HE | GC | **Lat.** |
| CIFAR-100 · SENets | | 300 | 2461 | 80.54 | 1004 | 33.7 | 1037 | 163 | 1055 | 80.86 | 770 | 18.4 | 788 | 1.8× | 2.3× | 0.3 | 1.3× | 1.8× | 1.3× |
| | | 240 | 2461 | 79.81 | 1004 | 27.0 | 1031 | 163 | 1055 | 80.86 | 770 | 18.4 | 788 | 1.5× | 2.3× | 1.1 | 1.3× | 1.5× | 1.3× |
| | | 180 | 2461 | 79.12 | 1004 | 20.2 | 1024 | 163 | 1055 | 80.86 | 770 | 18.4 | 788 | 1.1× | 2.3× | 1.7 | 1.3× | 1.1× | 1.3 × |
| | | 50 | 559 | 75.28 | 268 | 5.6 | 274 | 52 | 179 | 77.14 | 123 | 5.9 | 129 | 1.0× | 3.1× | 1.9 | 2.2× | 0.9× | 2.1× |
| | | 25 | 559 | 70.59 | 268 | 2.8 | 271 | 26 | 45 | 72.65 | 49 | 2.9 | 52 | 0.9× | 12.5× | 2.1 | 5.5× | 1.0× | **5.2×** |
| CIFAR-100 · SNL | | 15 | 559 | 67.17 | 268 | 1.7 | 270 | 13 | 179 | 68.25 | 123 | 1.5 | 124 | 1.1× | 3.1× | 1.1 | 2.2× | 1.1× | 2.2× |
| | | 13 | 559 | 66.53 | 268 | 1.5 | 270 | 13 | 179 | 68.25 | 123 | 1.5 | 124 | 1.0× | 3.1× | 1.7 | 2.2× | 1.0× | 2.2× |
| TinyImageNet · SENets | | 300 | 2227 | 64.96 | 927 | 33.7 | 961 | 327 | 1055 | 64.92 | 526 | 36.7 | 563 | 0.9× | 2.1× | 0.0 | 1.8× | 0.9× | 1.7× |
| | | 142 | 2227 | 58.90 | 927 | 16.0 | 943 | 104 | 179 | 58.90 | 97 | 11.7 | 108 | 1.4× | 12.4× | 0.0 | 9.6× | 1.4× | **8.7×** |
| TinyImageNet · SNL | | 489 | 9830 | 64.42 | 3690 | 55.0 | 3745 | 653 | 4216 | 67.58 | 2029 | 73.4 | 2102 | 0.7× | 2.3× | 3.2 | 1.8× | 0.7× | 1.8× |
| | | 489 | 9830 | 64.42 | 3690 | 55.0 | 3745 | 418 | 2842 | 66.10 | 1307 | 45.0 | 1352 | 1.2× | 3.5× | 1.7 | 2.8× | 1.2× | 2.8× |
| | | 298 | 2227 | 64.04 | 927 | 33.5 | 961 | 327 | 1055 | 64.92 | 526 | 36.7 | 563 | 0.9× | 2.1× | 0.9 | 1.8 × | 0.9× | 1.7× |
| | | 100 | 2227 | 58.94 | 927 | 11.2 | 939 | 104 | 179 | 58.90 | 97 | 11.7 | 108 | 1.0× | 12.4× | 0.0 | 9.6× | 1.0× | **8.7×** |
| | | 59 | 2227 | 54.40 | 927 | 6.6 | 934 | 52 | 712 | 54.46 | 329 | 5.9 | 335 | 1.1× | 3.1× | 0.1 | 2.8× | 1.1× | 2.8× |

arises from linear operations (convolution and fully-connected layers), and Garbled-circuit (GC) latency (Ball et al., 2019), resulting from ReLU computation. See the experiential setup details in Appendix J.

On CIFAR-100, SENet requires 300K ReLUs and 2461M FLOPs to reach 80.54% accuracy, whereas HRN-5x5x3x achieves 80.86% accuracy with only 163K ReLUs and 1055M FLOPs, providing 1.8× ReLU and 2.3× FLOPs saving. Similarly, at 25K ReLUs, our approach achieves a 2.1% accuracy gain with 12.5× FLOP reduction, thereby saving 5.2× runtime. Even at an extremely low ReLU count of 13K, HRN is 1.7% more accurate and achieves 2.2× runtime saving, compared to the SNL.

On TinyImageNet, HybReNets outperform SENet at both 300K and 142K ReLUs, improving runtime by 1.7× and 8.7×, respectively. Compared to SNL at 489K ReLUs, HybReNets are 3.2% (1.7%) more accurate with a 1.8× (2.8×) reduction in runtime. At lower ReLU counts of 100K and 59K, HybReNets match the accuracy with SNL and achieve a 12.4× and 3.1× FLOP reduction, which results in 8.7× and 2.8× runtime improvement, respectively.

Our primary insight from Table 7 is that FLOP reduction does not inherently guarantee a proportional reduction in HE latency, whereas a direct correlation exists between ReLU reduction and GC latency savings. In particular, a ∼12.5× FLOP reduction translates to 5.2× and 8.7× latency reduction on CIFAR-100 and TinyImageNet, respectively. This is due to the fact HE latency has an intricate dependency on the input/output packing (Aharoni et al., 2023), rotational complexity (Lou et al., 2020b;a; Huang et al., 2022) and slot utilization (Lee et al., 2022). We refer the readers to Juvekar et al. (2018) for details.

**Generality case study on ResNet34** We select ResNet34 for the DeepReShape generality study for two key reasons: (1) its consistent use for the case study in prior PI-specific network optimization studies (Jha et al., 2021; Cho et al., 2022b; Kundu et al., 2023), and (2) its stage compute ratio ($\phi_1=3$, $\phi_2=4$, $\phi_3=6$, and $\phi_4=3$) distinguishes it from ResNet18, results in different sets of HRN networks, HRN-4x6x3x and HRN-4x9x2x, upon applying Algorithm 1. We use HRN-4x6x3x for comparison with SOTA in Table 8. Network configuration and ReLU optimization details are presented in Table 14.

HybReNet advances the ReLU-accuracy Pareto on both CIFAR-100 and TinyImageNet, shown in Figures 11 (a, b). Table 8 quantifies the FLOPs-ReLU-Accuracy benefits and runtime savings. On CIFAR-100, compared to SOTA, HybReNet improves runtime by 3.1× with a significant gain in accuracy—9.8%, 7.2%, 5.9%, and 2.1% at 15K, 25K, 30K and 50K ReLUs (respectively). Further on TinyImageNet, SNL requires 300K ReLUs and 4646M FLOPs to reach 64% accuracy, whereas HybReNet matches this accuracy with 8.8× fewer FLOPs, leading to a runtime improvement of 6.3×. Conclusively, it highlights the effectiveness of DeepReShape and validates its generality for different network configurations and datasets.

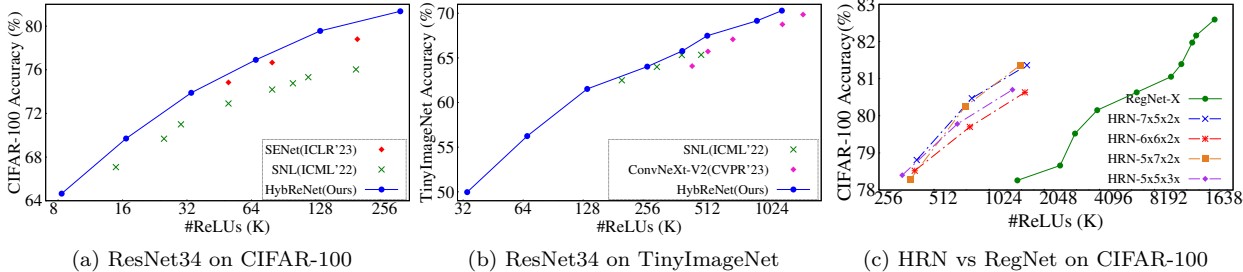

(a) ResNet34 on CIFAR-100     (b) ResNet34 on TinyImageNet     (c) HRN vs RegNet on CIFAR-100

Figure 11: HybReNets outperform SOTA ReLU-optimization methods applied to ResNet34 and also surpass SOTA FLOPs efficient models: RegNets and ConvNeXt-V2 (See Table 14 for the Pareto points specifics.).

Table 8: ResNet34-based HybReNets outperform SOTA PI methods (Kundu et al., 2023; Cho et al., 2022b) employed on ResNet34, and also surpass the SOTA FLOPs efficient models ConvNeXt-V2 (Woo et al., 2023). #Re and #FL denote ReLU and FLOPs counts; Acc. is top-1 accuracy; Lat. is the runtime for one PI.

| | | SOTA in Private Inference (on ResNet34) | | | | | | HybReNet(**Ours**) | | | | | | Improvements | | | | | |
|---|---|---|---|---|---|---|---|---|---|---|---|---|---|---|---|---|---|---|---|
| | | #Re | #FL | Acc. | HE | GC | Lat. | #Re | #FL | Acc. | HE | GC | Lat. | #Re | #FL | Acc. | HE | GC | **Lat.** |
| CIFAR-100 | SENet | 200 | 1162 | 78.80 | 459 | 22.5 | 482 | 134 | 527 | 79.56 | 404 | 15.1 | 419 | 1.5× | 2.2 × | 0.8 | 1.1× | 1.5× | 1.1× |
| | | 80 | 1162 | 76.66 | 459 | 9.0 | 468 | 67 | 132 | 76.91 | 140 | 7.5 | 148 | 1.2× | 8.8× | 0.3 | 3.3× | 1.2× | 3.2× |
| | | 50 | 1162 | 74.84 | 459 | 5.6 | 465 | 67 | 132 | 76.91 | 140 | 7.5 | 148 | 0.7× | 8.8× | 2.1 | 3.3× | 0.7× | 3.1× |
| | SNL | 30 | 1162 | 71.00 | 459 | 3.4 | 462 | 67 | 132 | 76.91 | 140 | 7.5 | 148 | 0.4× | 8.8× | **5.9** | 3.3× | 0.4× | **3.1×** |
| | | 25 | 1162 | 69.68 | 459 | 2.8 | 462 | 67 | 132 | 76.91 | 140 | 7.5 | 148 | 0.4× | 8.8× | **7.2** | 3.3× | 0.4× | **3.1×** |
| | | 15 | 1162 | 67.08 | 459 | 1.7 | 461 | 67 | 132 | 76.91 | 140 | 7.5 | 148 | 0.2× | 8.8× | **9.8** | 3.3× | 0.2× | **3.1×** |
| TinyImageNet | SNL | 500 | 4646 | 65.34 | 1710 | 56.2 | 1766 | 537 | 2109 | 67.48 | 880 | 60.3 | 940 | 0.9× | 2.2× | 2.1 | 1.9× | 0.9× | 2.3× |
| | | 400 | 4646 | 65.32 | 1710 | 45.0 | 1755 | 537 | 2109 | 67.48 | 880 | 60.3 | 940 | 0.7× | 2.2× | 2.2 | 1.9× | 0.7× | 2.3× |
| | | 300 | 4646 | 63.99 | 1710 | 33.7 | 1744 | 268 | 529 | 64.02 | 245 | 30.2 | 275 | 1.1× | 8.8× | 0.0 | 7.0× | 1.1× | **6.3×** |
| | | 200 | 4646 | 62.49 | 1710 | 22.5 | 1733 | 268 | 529 | 64.02 | 245 | 30.2 | 275 | 0.7× | 8.8× | 1.5 | 7.0× | 0.7× | **6.3×** |
| | ConvNeXt | 1622 | 11801 | 69.85 | 4067 | 182.4 | 4249 | 1270 | 8244 | 70.29 | 3091 | 142.8 | 3233 | 1.3× | 1.4× | 0.4 | 1.3× | 1.3× | 1.3× |
| | | 1278 | 9080 | 68.75 | 2368 | 143.7 | 2512 | 952 | 4638 | 69.15 | 1837 | 107.1 | 1944 | 1.3× | 2.0× | 0.4 | 1.3× | 1.3× | 1.3× |
| | | 721 | 3436 | 67.08 | 1307 | 81.0 | 1388 | 537 | 2109 | 67.48 | 880 | 60.3 | 940 | 1.3× | 1.6× | 0.4 | 1.5× | 1.3× | 1.5× |
| | | 541 | 1935 | 65.72 | 738 | 60.8 | 799 | 402 | 1187 | 65.77 | 592 | 45.2 | 637 | 1.3× | 1.6× | 0.0 | 1.3× | 1.3× | 1.3× |
| | | 451 | 1345 | 64.07 | 546 | 50.7 | 597 | 268 | 529 | 64.02 | 245 | 30.2 | 275 | 1.7× | 2.5× | 0.0 | 2.2× | 1.7× | 2.2× |

**HybReNet outperform SOTA FLOPs efficient vision models** We perform a comparative analysis of HybReNets with SOTA FLOPs efficient vision models: ConvNeXt-V2 (Woo et al., 2023) and RegNet (Radosavovic et al., 2020). These models possess distinct depth and width hyperparameters, providing an interesting case study, particularly when contrasted with conventional ReNets. See Appendix H.4 for details.

For a fair comparison with baseline RegNet-X models, we do not employ any ReLU-optimization steps on (ResNet18-based) HybReNets. Results are shown in Figure 11(c) where HRNs are evaluated with $m \in \{16, 32, 64\}$. HRNs achieve comparable accuracy with substantially fewer ReLUs compared to RegNets. For instance, to achieve 78.26% (80.63%) accuracy on CIFAR-100, RegNets require 1460K (6544K) ReLUs, while HRN-5x5x3x needs only 343K (1372K) ReLUs, leading to a 4.3× (4.7×) ReLU reduction.

Further, we compare the ConvNeXt-V2 models with HybReNets on TinyImageNet while employing ReLU optimization on them (see Table 14 for optimization details). The ReLU-accuracy Pareto is shown in Figure 11(b), with a detailed comparison outlined in Table 8. The competing HRNs achieve 1.3× to 1.7× ReLU savings; 1.4× to 2.5× FLOP reduction, which results in 1.3× to 2.3× runtime improvements.

**ReLU-reuse is more effective for HybReNets and outperforms the SOTA channel-wise ReLU optimization** We examine the efficacy of ReLU-reuse on networks with various ReLUs' distributions and compare their performance with conventional (channel/feature-map)scaling used in DeepReDuce for achieving very low ReLU counts. Results are shown in Figure 22 and Figure 23 (AppendixI). Interestingly, we observed that the efficacy of ReLU-reuse is most pronounced in networks where ReLUs are aligned in their criticality order, whether partially or entirely. In fact, networks with an even distribution of stagewise ReLUs exhibit more significant accuracy improvements from ReLU-reuse compared to traditional networks like ResNets.

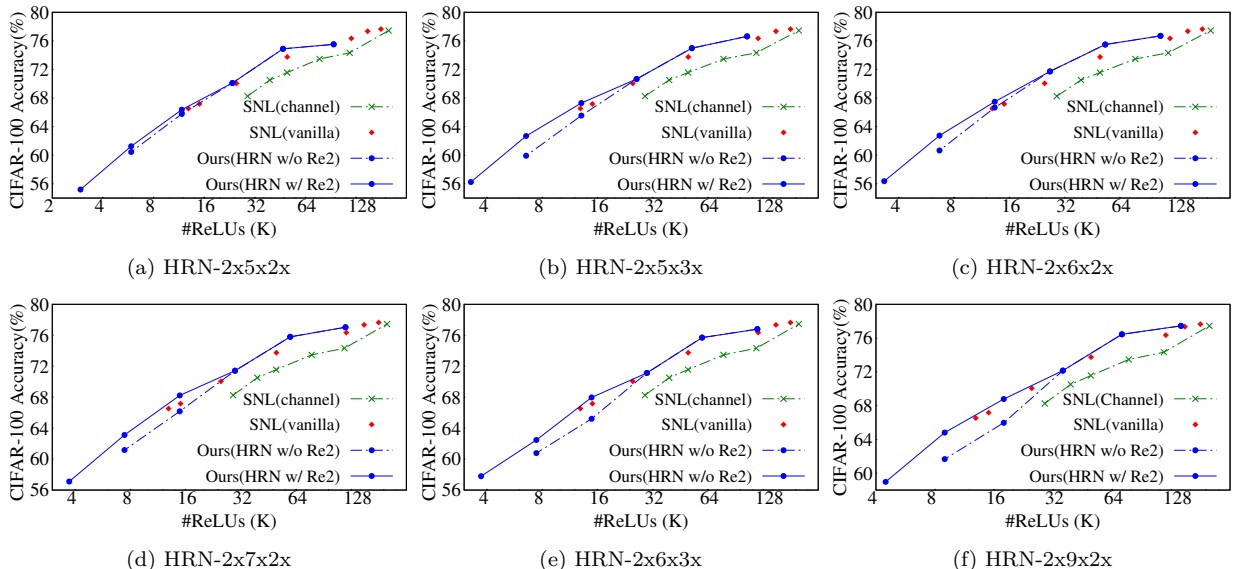

(a) HRN-2x5x2x     (b) HRN-2x5x3x     (c) HRN-2x6x2x

(d) HRN-2x7x2x     (e) HRN-2x6x3x     (f) HRN-2x9x2x

Figure 12: ReLU-reuse (Re2) consistently outperforms the SOTA channel-wise ReLU dropping technique used in SNL across various ReLU counts. Substituting the conventional scaling method used in DeepReDuce (denoted as "w/o Re2") with Re2 results in an accuracy gain of **1% - 3%**, bringing the performance closer to the pixel-wise SNL (denoted as "SNL(vanilla)").

Further, we employ ReLU-reuse on HRNs with $\alpha=2$, as per Algorithm 2, and compare their performance with SOTA channel-wise ReLU optimization method used in SNL. For a fair comparison, we use standard knowledge distillation (Hinton et al., 2015), as used in SNL[5], rather than DKD (Zhao et al., 2022). Figure 12 demonstrates that Re2 results in a significant accuracy improvement of up to **3%**. This gain in accuracy enables HRNs to achieve performance on par with pixel-wise SNL.

**Ablation study for ReLU-reuse** We conduct an ablation study on ResNet18 to investigate the benefits of two key techniques employed in ReLU-reuse: (1) shortcut connections between outputs and inputs of subsequent feature-subspaces (see Figure 9), and (2) using a fixed number of divisions in feature maps regardless of the ReLU reduction factor. We removed ReLUs from alternate layers using ReLU-Thinning and integrated ReLU-reuse in the others, and results are shown in Table 9. The results show that shortcut connections boost accuracy at lower ReLU reduction factors, but their benefit diminishes with higher reduction factors. Specifically, accuracy drops by 1.5% when the reduction factor increases from 2× to 4×. This reduction is likely due to the significant loss of cross-channel information with more divisions in feature-map.

Table 9: Results for an ablation study where ReLU-reuse is employed in alternate convolution layers in (i.e., ReLU-Thinned) ResNet18 (CIFAR-100). The constant number of divisions (i.e., 3) in the proposed approach of ReLU-reduction *offers scalability for higher ReLU reduction factors.* The term *reuse* in the table refers to shortcut connections between feature-subspaces in $N$ partitions (see Figure 9).

| ReLU-reduction factor | #ReLUs | N divisions | | **Proposed** |
|---|---|---|---|---|
| | | w/o Reuse | w/ Reuse | (3 divisions) |
| 2x ReLU reduction ($N=2$) | 434.18K | 77.61% | 78.19% | 77.83% |
| 4x ReLU reduction ($N=4$) | 372.74K | 75.84% | 76.87% | 77.60% |
| 8x ReLU reduction ($N=8$) | 342.02K | 75.43% | 75.66% | 76.93% |
| 16x ReLU reduction ($N=16$) | 326.66K | 75.33% | 75.47% | 76.38% |

---

[5]It is important to note that SENets (Kundu et al., 2023) uses PRAM (Post-ReLU Activation Mismatch) loss in conjunction with standard KD (Hinton et al., 2015) for an additional boost in the accuracy of ReLU-reduced models. In contrast, both SNL (Cho et al., 2022b) and DeepReDuce (Jha et al., 2021) rely solely on standard KD.

On the other hand, a fixed number of divisions in our proposed approach stabilizes the accuracy degradation even at higher ReLU reduction factors, emphasizing their scalability for achieving significantly lower ReLU reductions. Note that, at a reduction factor of 2, the ReLU-reuse technique demonstrated slightly lower accuracy than the $N$ division method with shortcut connections. This is because the latter consists of only two groups of feature maps, while the former has three, which resulted in more information loss.

**The baseline HybReNets exhibits superior ReLU efficiency compared to the standard networks used in private inference**

We evaluated the ReLU efficiency of baseline HRNs without leveraging any coarse or fine-grained ReLU optimization methods, as well as knowledge distillation. We compared them with two widely used network architectures in PI: ResNet and WideResNets. Results are shown in Figure 13. The homogeneous channel scaling in ResNet18 StageCh networks led to superior ReLU efficiency than WideResNets variants until accuracy in the former is saturated. Nonetheless, all the four HRNs—HRN-5x5x3x, HRN-5x7x2x, HRN-6x6x2x, and HRN-7x5x2x—exceeds the ReLU efficiency of ResNet18 StageCh variants, demonstrating the benefits of strategically allocating channels in the subsequent stages of the classical networks for PI.

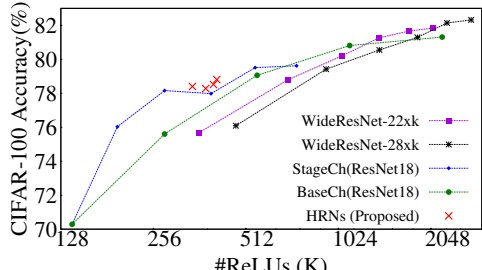

Figure 13: ReLU efficiency comparison with baseline HybReNets. For WideResNets $k \in \{2, 4, 6, 8, 10, 12\}$.

# 6 Related Work

**PI-specific network optimization** Delphi (Mishra et al., 2020), SAFENet (Lou et al., 2021), and Garimella et al. (2021) substitute ReLUs with low-degree polynomials, while AutoFHE (Ao & Boddeti, 2024) performed layerwise mixed-degree polynomial substitution. Ghodsi et al. (2021) proposed stochastic ReLU, a probabilistic approximation of ReLU functions, and co-optimized the garbled circuits. DeepReDuce (Jha et al., 2021), a manual coarse-grained ReLU optimization method, drops ReLUs layerwise. SNL (Cho et al., 2022b) and SENet (Kundu et al., 2023) are fine-grained ReLU optimization and drop the pixel-wise ReLUs. CryptoNAS (Ghodsi et al., 2020) and Sphynx (Cho et al., 2022a) use neural architecture search and employ a constant number of ReLUs per layer for designing ReLU-efficient networks, disregarding FLOPs implications. In contrast, our approach achieves ReLU and FLOP efficiency simultaneously. We refer the reader to Ng & Chow (2023) for detailed HE and GC-specific optimizations for private inference. A recent work Zeng et al. (2023b) used oblivious transfer for nonlinear operations and rotation-free homomorphic encryption (Huang et al., 2022) for linear layers, and showed that communication cost is dominated by linear operations.

**Benefits of width** The impact of network width on reducing catastrophic forgetting was highlighted by Mirzadeh et al. (2022). The influence of network width on the smoothness of the loss surface was analyzed by Li et al. (2018), and it was found that an increase in width could mitigate erratic behavior in the loss landscape. A study by Golubeva et al. (2021) decoupled the effects of increased width from over-parameterization and found that the width of a network primarily determines its predictive performance, with the number of parameters being a secondary factor under mild assumptions. Nguyen et al. (2021) established that wider networks, when delivering similar levels of accuracy on the ImageNet dataset, show superior performance on inputs that reflect the scene rather than the objects.

**Challenges and implications of nonlinear layers in diverse neural network applications** Nonlinear layers not only present challenges in private inference; they introduce significant hurdles across various neural network applications. For instance, in the realm of optical neural networks, ReLUs exacerbate energy consumption and increase latency due to the costs associated with optical-to-electrical signal conversions, which in turn diminishes the overall system efficiency (Chang et al., 2018; Li et al., 2022). When it comes to verifying adversarial robustness, the prevalence of ReLUs can make the process notably more time-intensive. This increase in complexity arises from the higher proportion of unstable neurons (Xiao et al., 2019; Balunović & Vechev, 2020; Chen et al., 2022a).

Additionally, ReLUs considerably hinder the progress of verifiable machine learning because its non-arithmetic operations are incompatible with zero-knowledge proof systems (Sun & Zhang, 2023), and prior work has

resorted to employing polynomial approximations (Ali et al., 2020; Zhao et al., 2021; Eisenhofer et al., 2022) or have implemented methods based on lookup tables (Liu et al., 2021; Kang et al., 2022). Furthermore, the non-distributive nature of ReLU over rotation operations can break the equivariance property of Steerable CNNs (Franzen & Wand, 2021), known for their parameter and computation efficiency (Cohen & Welling, 2017; Weiler et al., 2018; Weiler & Cesa, 2019); thus, limiting their architectural choices and applicability.

Thus, the ReLU optimization techniques of DeepReShape not only address the challenges in private inference but also hold promise for broader applications, suggesting its versatility and potential for widespread impact.

## 7 Discussion

**Conclusion and broader impact** Privacy-preserving computations demand substantial resources, particularly in terms of storage, communication bandwidth, and compute power. Using the garbled-circuit technique alone can consume hundreds of gigabytes of storage, while homomorphic computations might need hours to complete a single private inference in real-world scenarios (Rathee et al., 2020; Garimella et al., 2023). Researcher have proposed specialized hardware accelerators (Samardzic et al., 2021; 2022; Soni et al., 2023; Mo et al., 2023; Kim et al., 2023; Agrawal et al., 2024; Putra et al., 2024) and (cryptographic) protocol improvements to tackle these challenges. Yet, these solutions present challenges of their own: hardware solutions may not always be sustainable in the long run (Gupta et al., 2022), and protocol tweaks could potentially open doors to security vulnerabilities or raise compatibility concerns.

In this context, our research shifts the focus towards algorithmic innovations and aims to address the unique challenge of reducing FLOPs without compromising ReLU efficiency. We proposed DeepReShape to optimize FLOP count while maintaining ReLU efficiency effectively. We achieve this by identifying superfluous FLOPs in conventional ReLU efficient networks and understanding that wide networks are mainly beneficial for higher ReLU counts, providing additional opportunities for FLOP reduction when targeting lower ReLU counts. By leveraging these insights, we achieve FLOP reduction up to $45\times$ without any bells and whistles.

One significant advantage of algorithmic improvement is their adaptability across diverse hardware configurations and cryptographic protocols, thus broadening the potential impact of our algorithmic innovations. We showed that a substantial reduction in runtime, $\sim(5\times$ to $10\times)$, can be achieved simply by strategically allocating channels in the existing networks and employing straightforward ReLU optimization steps.

Furthermore, as discussed in §6, nonlinear layers are a bottleneck also in other areas of machine learning privacy and security. Thus, our work on the simultaneous optimization of ReLU and FLOPs holds promise for broader applications in these fields.

**Limitations** Achieving a specific target ReLU count with HybReNets is challenging due to the coarse-grained nature of ReLU optimization steps. Fine-grained optimization leads to suboptimal performance in HybReNets because of changes in the ReLUs' distribution compared to conventional CNNs. Coarse-grained ReLU optimization steps either halve the network's ReLU count or remove all ReLUs from Stage1, depending on the ReLU distribution in the network (see Algorithm 2). Consequently, the final ReLU count depends on the baseline network's initial ReLU count and their distribution within the network, influenced by the base channels and stage-wise channel multiplication factors.

**Future work** Developing PI-efficient networks from scratch could yield more optimized networks for PI performance; however, it requires exhaustive design space exploration and the training of multiple subnetworks to successively narrow the search space. This makes it computationally intensive process. Nonetheless, there is a significant potential for creating families of networks tailored for optimal PI performance. Furthermore, additional reductions in FLOPs can be achieved by employing techniques such as linear layer fusion, as demonstrated in (Jha et al., 2021; Dror et al., 2021; Zeng et al., 2023a).

## Acknowledgment

We would like to thank Karthik Garimella for his assistance in computing the runtime (HE and GC latency) for private inference. This research was developed with funding from the Defense Advanced Research

Projects Agency (DARPA), under the Data Protection in Virtual Environments (DPRIVE) program, contract HR0011-21-9-0003. The views, opinions and/or findings expressed are those of the author and should not be interpreted as representing the official views or policies of the Department of Defense or the U.S. Government.

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

# Appendix

## Table of Contents

## A    Design Rationale for Hyper-Parameter Selection in HybReNet Networks

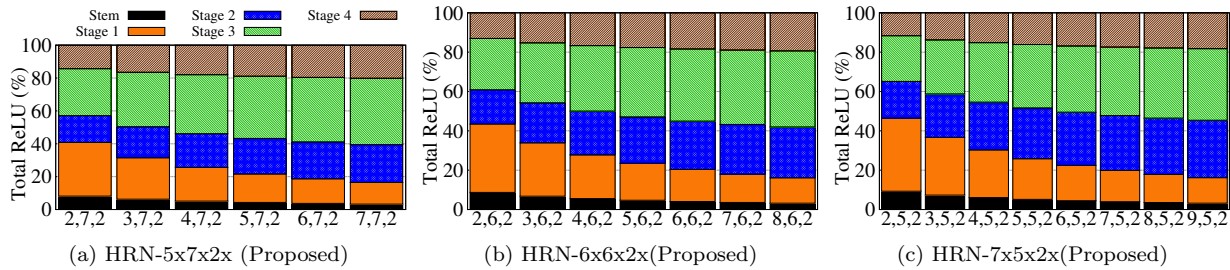

Figure 14: Analyzing ReLUs' distribution in HRNs by progressively increasing the $\alpha$ values from $\alpha=2$. Once the network achieves ReLU equalization—(5, 7, 2) for HRN-5x7x2x, (6, 6, 2) for HRN-6x6x2x, and (7, 5, 2) for HRN-7x5x2x— the ReLUs' distribution remains stable with increasing $\alpha$ value.

In this section, we explain our design decisions for choosing specific $\alpha$, $\beta$, and $\gamma$ in HybReNets. We selected the smallest $\alpha$ within a specified range for the given pairs of $(\beta, \gamma)$ based on two primary considerations

Firstly, when the network attains ReLU equalization, the ReLU distribution becomes stable and remains constant as $\alpha$ grows. This stability is due to the fact that altering $\alpha$ has the least impact on the relative distribution of stagewise ReLUs compared to increasing $\beta$ and $\gamma$ (Figure 14). Specifically, increasing $\alpha$ results in a slight decrease in the proportion of Stage 1 ReLUs and a slight increase in the remaining stages.

Secondly, when ReLU optimization (Jha et al., 2021) is employed, increasing $\alpha$ in HRNs does not improve ReLU efficiency. Instead, it results in an inferior ReLU-accuracy tradeoff at lower ReLU counts (Figure 15).

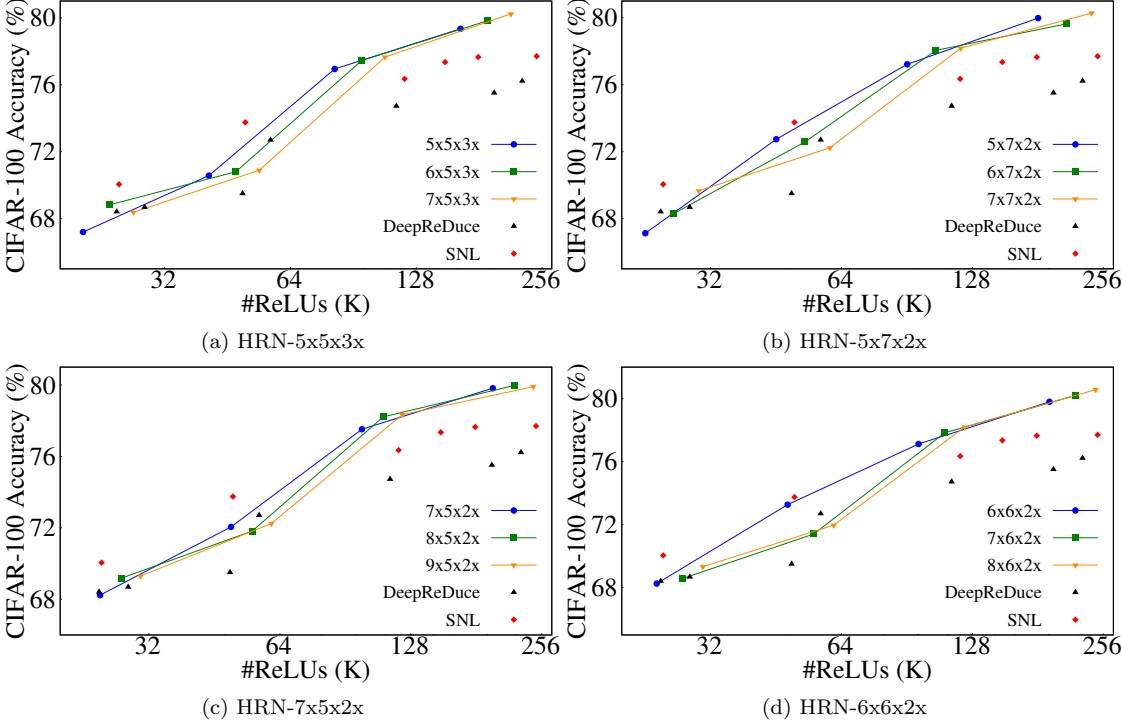

Figure 15: Effect of increasing $\alpha$ in HybReNets: The ReLU efficiency of networks with higher $\alpha$ does not improve; in fact, it significantly reduces at lower ReLU counts.

# B ReLUs' Criticality Order in StageCh, BaseCh and HybReNet Networks

Table 10: Evaluating stage-wise ReLU criticality in ResNet18 (R18) BaseCh and StageCh networks on CIFAR-100. Criticality metrics ($C_k$) are determined using the method from Jha et al. (2021). Both BaseCh and StageCh networks maintain the original ResNet18 criticality order: $S_3 > S_2 > S_4 > S_1$ (Higher $C_k$ implies more critical ReLUs).

| Networks | Stage1 | | | | Stage2 | | | | Stage3 | | | | Stage4 | | | |
|---|---|---|---|---|---|---|---|---|---|---|---|---|---|---|---|---|
| | #ReLUs | Acc(%) | +KD(%) | $C_k$ | #ReLUs | Acc(%) | +KD(%) | $C_k$ | #ReLUs | Acc(%) | +KD(%) | $C_k$ | #ReLUs | Acc(%) | +KD(%) | $C_k$ |
| R18(m=16)-2x2x2x | 81.92K | 52.08 | 52.67 | 0.00 | 32.77K | 61.24 | 62.10 | 7.39 | 16.38K | 63.00 | 64.64 | 9.84 | 8.19K | 58.09 | 59.70 | 6.07 |
| R18(m=32)-2x2x2x | 163.84K | 59.19 | 60.19 | 0.00 | 65.54K | 65.91 | 66.47 | 4.69 | 32.77K | 65.7 | 67.28 | 5.55 | 16.38K | 60.48 | 62.22 | 1.67 |
| R18(m=64)-2x2x2x | 327.68K | 62.65 | 63.13 | 0.00 | 131.07K | 67.18 | 68.32 | 3.69 | 65.54K | 68.75 | 70.29 | 5.34 | 32.77K | 62.63 | 63.47 | 0.27 |
| R18(m=128)-2x2x2x | 655.36K | 62.34 | 64.15 | 0.00 | 262.14K | 69.28 | 70.56 | 4.34 | 131.07K | 71.25 | 72.04 | 5.61 | 65.54K | 63.59 | 64.58 | 0.32 |
| R18(m=256)-2x2x2x | 1310.72K | 64.81 | 65.22 | 0.00 | 524.29K | 71.95 | 72.43 | 4.65 | 262.14K | 72.69 | 73.77 | 5.79 | 131.07K | 64.79 | 65.77 | 0.39 |
| R18(m=16)-3x3x3x | 81.92K | 52.77 | 53.07 | 0.00 | 49.15K | 64.93 | 65.67 | 9.59 | 36.86K | 66.23 | 67.96 | 11.57 | 27.65K | 61.74 | 63.43 | 8.21 |
| R18(m=16)-4x4x4x | 81.92K | 52.19 | 52.20 | 0.00 | 65.54K | 65.62 | 66.22 | 10.46 | 65.54K | 67.82 | 69.16 | 12.66 | 65.54K | 63.52 | 65.46 | 9.89 |
| R18(m=16)-5x5x5x | 81.92K | 50.38 | 50.65 | 0.00 | 81.92K | 66.10 | 66.63 | 11.74 | 102.40K | 70.17 | 70.64 | 14.46 | 128.00K | 64.86 | 65.43 | 10.52 |
| R18(m=16)-6x6x6x | 81.92K | 50.60 | 51.53 | 0.00 | 98.30K | 66.74 | 67.11 | 11.30 | 147.46K | 70.67 | 72.09 | 14.49 | 221.18K | 65.22 | 66.43 | 10.21 |
| R18(m=16)-7x7x7x | 81.92K | 50.93 | 49.07 | 0.00 | 114.69K | 66.59 | 67.89 | 13.50 | 200.70K | 72.08 | 73.33 | 16.74 | 351.23K | 65.95 | 67.88 | 12.48 |

It remains intriguing to examine whether the ReLUs' criticality order in baseline networks, such as ResNet18, remains consistent when the network width is modified, specifically in the BaseCh, StageCh, and HRN variations. To explore this, we computed the stagewise criticality metric for ResNet18 BaseCh and StageCh networks (Table 10), and HRN networks with $\alpha$ values between 2 and 7 (Table 11). Interestingly, the criticality order of the standard ResNet18 remains preserved in BaseCh and StageCh models, as well as in all HRNs, except for those with $\alpha$=2 (HRN-2x5x3x, HRN-2x5x2x, HRN-2x6x2x, and HRN-2x7x2x). Specifically, in HRNs with $\alpha$=2, the criticality order of Stage2 and Stage3 is shuffled, while the most and least critical stages remain unchanged (i.e., $S_3 > S_2 > S_4 > S_1$). To account for this altered criticality order, we recomputed $\alpha$, $\beta$, and $\gamma$ using Algorithm 1, resulting in two HRNs: HRN-2x6x3x and HRN-2x9x2x. However, the criticality order in these two HRNs did not adapt to the altered criticality order (highlighted in green in Table 11).

Table 11: Evaluating stage-wise ReLU criticality in ResNet18-based HRN networks with $\alpha$ values from 2 to 7 on CIFAR-100. Criticality metrics ($C_k$) for each stage are determined using the method in Jha et al. (2021). Except for $\alpha$=2, all HRN networks maintain the original ResNet18 criticality order ($S_3 > S_2 > S_4 > S_1$). HRNs with the minimum $\alpha$, $\beta$, and $\gamma$ required for full ReLU equalization are highlighted in gray. The HRNs highlighted in green are designed for a different criticality order: $S_3 > S_4 > S_2 > S_1$.

| Networks | Stage1 | | | | Stage2 | | | | Stage3 | | | | Stage4 | | | |
|---|---|---|---|---|---|---|---|---|---|---|---|---|---|---|---|---|
| | #ReLUs | Acc(%) | +KD(%) | $C_k$ | #ReLUs | Acc(%) | +KD(%) | $C_k$ | #ReLUs | Acc(%) | +KD(%) | $C_k$ | #ReLUs | Acc(%) | +KD(%) | $C_k$ |
| HRN-2x7x2x | 81.92K | 52.14 | 53.39 | 0.00 | 32.77K | 61.63 | 61.59 | 6.42 | 57.34K | 68.44 | 69.82 | 12.37 | 28.67K | 62.15 | 63.40 | 7.91 |
| HRN-3x7x2x | 81.92K | 51.61 | 53.29 | 0.00 | 49.15K | 64.46 | 65.26 | 9.11 | 86.02K | 69.88 | 70.77 | 12.80 | 43.01K | 63.10 | 64.17 | 8.36 |
| HRN-4x7x2x | 81.92K | 51.28 | 49.42 | 0.00 | 65.54k | 65.93 | 66.47 | 12.72 | 114.69K | 70.94 | 72.16 | 16.32 | 57.34K | 63.70 | 64.77 | 11.56 |
| HRN-5x7x2x | 81.92K | 49.82 | 48.36 | 0.00 | 81.92K | 66.17 | 67.59 | 14.13 | 143.36K | 71.40 | 72.18 | 16.83 | 71.68K | 64.10 | 65.35 | 12.60 |
| HRN-6x7x2x | 81.92K | 51.23 | 48.48 | 0.00 | 98.30K | 66.88 | 68.06 | 14.20 | 172.03K | 71.86 | 72.73 | 16.91 | 86.02K | 64.15 | 65.75 | 12.64 |
| HRN-7x7x2x | 81.92K | 50.11 | 52.40 | 0.00 | 114.69K | 66.92 | 68.29 | 11.40 | 200.70K | 71.69 | 73.16 | 14.32 | 100.35K | 63.82 | 65.53 | 9.51 |
| HRN-2x6x2x | 81.92K | 52.29 | 53.19 | 0.00 | 32.77K | 61.62 | 62.00 | 6.90 | 49.15K | 67.36 | 69.51 | 12.43 | 24.58K | 61.64 | 63.25 | 8.04 |
| HRN-3x6x2x | 81.92K | 52.50 | 52.80 | 0.00 | 49.15K | 64.50 | 65.64 | 9.78 | 73.73K | 68.61 | 70.96 | 13.44 | 36.86K | 62.77 | 64.09 | 8.77 |
| HRN-4x6x2x | 81.92K | 53.23 | 53.32 | 0.00 | 65.54K | 65.74 | 66.03 | 9.48 | 98.30K | 70.47 | 71.54 | 13.22 | 49.15K | 63.59 | 64.82 | 8.76 |
| HRN-5x6x2x | 81.92K | 50.79 | 51.64 | 0.00 | 81.92K | 66.89 | 67.27 | 11.48 | 122.88K | 70.33 | 71.50 | 14.18 | 61.44K | 63.97 | 64.94 | 9.97 |
| HRN-6x6x2x | 81.92K | 50.01 | 50.59 | 0.00 | 98.30K | 66.57 | 67.94 | 12.58 | 147.46K | 71.18 | 72.59 | 15.51 | 73.73K | 64.13 | 65.39 | 10.95 |
| HRN-7x6x2x | 81.92K | 51.01 | 49.64 | 0.00 | 114.69K | 66.74 | 68.57 | 13.58 | 172.03K | 71.84 | 72.84 | 16.18 | 86.02K | 64.54 | 65.16 | 11.36 |
| HRN-2x5x2x | 81.92K | 52.03 | 53.05 | 0.00 | 32.77K | 61.60 | 61.76 | 6.82 | 40.96K | 66.64 | 68.29 | 11.75 | 20.48K | 61.02 | 62.58 | 7.71 |
| HRN-3x5x2x | 81.92K | 53.23 | 52.61 | 0.00 | 49.15K | 64.57 | 65.71 | 9.97 | 61.44K | 68.40 | 69.93 | 12.98 | 30.72K | 62.32 | 63.42 | 8.51 |
| HRN-4x5x2x | 81.92K | 52.65 | 52.33 | 0.00 | 65.54K | 65.60 | 66.89 | 10.86 | 81.92K | 69.81 | 70.85 | 13.61 | 40.96K | 63.14 | 63.94 | 8.95 |
| HRN-5x5x2x | 81.92K | 49.15 | 51.16 | 0.00 | 81.92K | 66.26 | 67.47 | 11.98 | 102.40K | 70.15 | 71.69 | 14.85 | 51.20K | 63.55 | 64.67 | 10.26 |
| HRN-6x5x2x | 81.92K | 49.06 | 52.10 | 0.00 | 98.30K | 66.56 | 68.08 | 11.59 | 122.88K | 71.33 | 71.85 | 14.10 | 61.44K | 63.59 | 64.89 | 9.59 |
| HRN-7x5x2x | 81.92K | 51.58 | 51.93 | 0.00 | 114.69K | 66.94 | 67.89 | 11.45 | 143.36K | 70.79 | 72.87 | 14.79 | 71.68K | 64.02 | 65.23 | 9.86 |
| HRN-2x5x3x | 81.92K | 52.36 | 53.68 | 0.00 | 32.77K | 61.39 | 61.30 | 5.97 | 40.96K | 66.78 | 68.17 | 11.17 | 30.72K | 62.01 | 63.83 | 7.99 |
| HRN-3x5x3x | 81.92K | 51.05 | 52.89 | 0.00 | 49.15K | 64.64 | 65.10 | 9.30 | 61.44K | 68.87 | 70.14 | 12.93 | 46.08K | 63.66 | 64.32 | 8.74 |
| HRN-4x5x3x | 81.92K | 51.57 | 50.62 | 0.00 | 65.54K | 65.66 | 66.06 | 11.52 | 81.92K | 69.12 | 70.13 | 14.33 | 61.44K | 63.64 | 65.58 | 11.21 |
| HRN-5x5x3x | 81.92K | 50.22 | 52.41 | 0.00 | 81.92K | 66.42 | 67.55 | 11.12 | 102.40K | 70.15 | 70.97 | 13.42 | 76.80K | 64.21 | 65.59 | 9.73 |
| HRN-6x5x3x | 81.92K | 50.28 | 50.45 | 0.00 | 98.30K | 65.95 | 67.61 | 12.45 | 122.88K | 70.68 | 71.29 | 14.88 | 92.16K | 64.37 | 65.87 | 11.23 |
| HRN-7x5x3x | 81.92K | 50.12 | 50.31 | 0.00 | 114.69K | 66.85 | 67.95 | 12.66 | 143.36K | 71.20 | 71.87 | 15.23 | 107.52K | 64.72 | 65.58 | 11.01 |
| HRN-2x9x2x | 81.92K | 51.86 | 53.22 | 0.00 | 32.77K | 61.13 | 61.65 | 6.60 | 73.73K | 69.46 | 70.28 | 12.63 | 36.86K | 62.53 | 64.25 | 8.57 |
| HRN-2x6x3x | 81.92K | 52.75 | 52.85 | 0.00 | 32.77K | 61.33 | 61.44 | 6.73 | 49.15K | 67.36 | 68.76 | 12.11 | 36.86K | 62.69 | 64.59 | 9.12 |

## C   Adapting HybReNet Design to Criticality Order Variations

We conducted an exhaustive characterization of HRN networks designed for the prevalent criticality order: Stage3 > Stage2 > Stage4 > Stage1. However, we observed that the criticality order of Stage2 and Stage4 can change in some instances, such as when using HRNs with $\alpha=2$ or when applying ResNet18/ResNet34 on TinyImageNet (Jha et al., 2021). In these cases, the criticality order shifts to Stage3 > Stage4 > Stage2 > Stage1. This raises the question of whether running the criticality test for every baseline network on different datasets is necessary.

To address this, we compared the ReLU-accuracy performance of HRN networks designed with two different criticality orders. Using the DeepReShape algorithm (Algorithm 1), we designed HybReNets for the alternative criticality order of Stage3 > Stage4 > Stage2 > Stage1.

$$\#ReLUs(S_3) > \#ReLUs(S_4) > \#ReLUs(S_2) > \#ReLUs(S_1)$$

$$\implies \phi_3\Big(\frac{\alpha\beta}{16}\Big) > \phi_4\Big(\frac{\alpha\beta\gamma}{64}\Big) > \phi_2\Big(\frac{\alpha}{4}\Big) > \phi_1$$

ReLU equalization through width $(\phi_1 = \phi_2 = \phi_3 = \phi_4 = 2, \text{ and } \alpha \geq 2, \beta \geq 2, \gamma \geq 2):$

$$\implies \frac{\alpha\beta}{16} > \frac{\alpha\beta\gamma}{64} > \frac{\alpha}{4} > 1 \implies \alpha\beta > 16, \alpha > 4, \alpha\beta\gamma > 64, \beta > 4, \beta\gamma > 16, \text{ and } \gamma < 4$$

Solving the above compound inequalities provides the following range of $\beta$ and $\gamma$ at two different $\gamma$

At $\gamma = 2,\ \beta > 8\ \&\ \alpha > 4;\ \text{and at } \gamma = 3,\ \beta > 5\ \&\ \alpha > 4$

Solving these inequalities provides the following ranges for $\beta$ and $\gamma$ at different $\gamma$ values:

- At $\gamma = 2$: $\beta > 8$ and $\alpha > 4$

- At $\gamma = 3$: $\beta > 5$ and $\alpha > 4$

HRNs with the minimum values of $\alpha$, $\beta$, and $\gamma$ that satisfy the ReLU equalization for the altered criticality order (Stage3 > Stage4 > Stage2 > Stage1) are HRN-5x6x3x and HRN-5x9x2x. For lower ReLU counts, we select HRNs with $\alpha=2$, (i.e., HRN-2x6x3x and HRN-2x9x2x). We compare the ReLU-accuracy tradeoffs of these HRNs with those designed for the prevalent criticality order using both coarse-grained ReLU optimization (DeepReDuce) and fine-grained ReLU optimization (SNL).

The results (Figure 16) show that the performance of HRNs for both criticality orders is similar with coarse-grained optimization on CIFAR-100. However, with fine-grained optimization, there is a noticeable accuracy gap. Specifically, HRN-2x5x3x and HRN-2x7x2x outperform HRN-2x6x3x and HRN-2x9x2x by a small but discernible margin. On TinyImageNet, HRN-5x6x3x and HRN-5x9x2x perform similarly, except that HRN-5x5x3x outperforms at some intermediate ReLU counts.

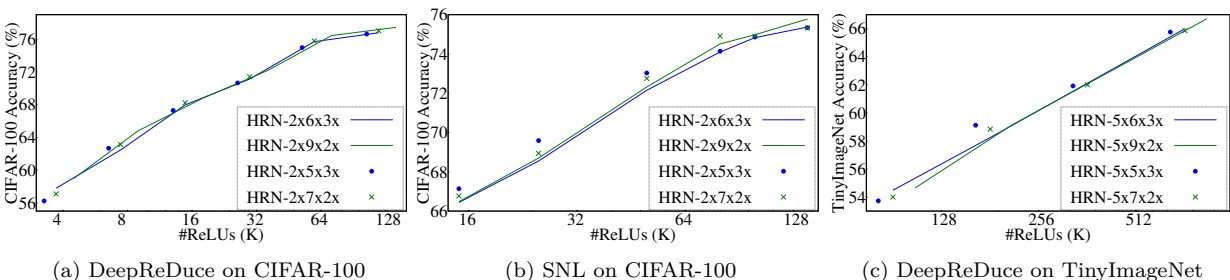

(a) DeepReDuce on CIFAR-100    (b) SNL on CIFAR-100    (c) DeepReDuce on TinyImageNet

Figure 16: Performance comparison of HRNs designed for the altered criticality order (Stage3 > Stage4 > Stage2 > Stage1) with $\beta=6$ & 9, and HRNs designed for the prevalent criticality order (Stage3 > Stage2 > Stage4 > Stage1) with $\beta=5$ & 7. Overall, the latter exhibit slightly better performance than the former.

# D Depth-Based ReLU Equalization

ReLU equalization through width in HybReNets has two effects: increasing the network's complexity per unit of nonlinearity (measured as parameters and FLOPs per unit of ReLU) and aligning the ReLU distribution according to their criticality order. To analyze these effects independently, we applied ReLU equalization through depth and augmented the base channel counts to increase parameters and FLOPs per unit of ReLU.

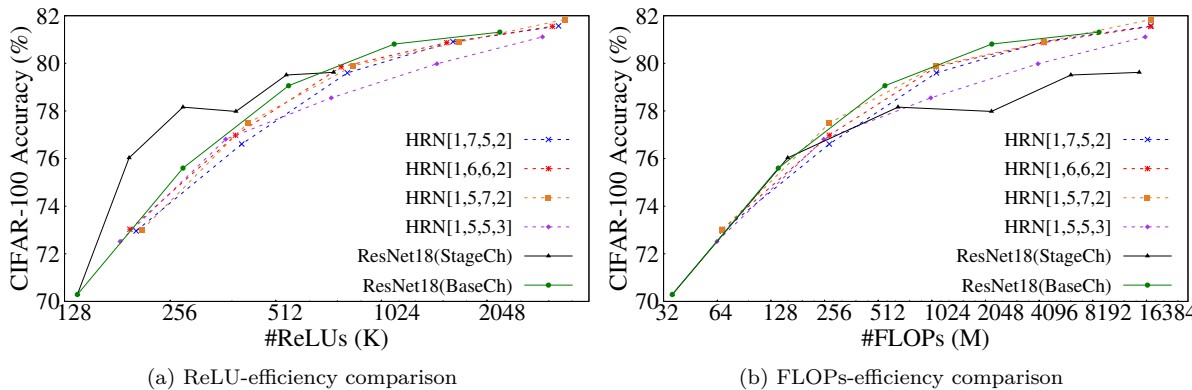

(a) ReLU-efficiency comparison          (b) FLOPs-efficiency comparison

Figure 17: Efficacy of depth-based ReLU equalization, performed using depth hyper-parameters, i.e., stage-compute-ratios (modified values shown in brackets $[\phi_1, \phi_2, \phi_3, \phi_4]$). These depth-based HRNs show similar or worse ReLU and FLOPs efficiency compared to *BaseCh* networks, highlighting the effectiveness of width adjustments through $\alpha$, $\beta$, and $\gamma$ for ReLU equalization in width-based HybReNets.

We use ResNet18 with $m{=}16$ and fixed $\alpha{=}\beta{=}\gamma{=}2$ while setting the stage-compute-ratios ($\phi_1$, $\phi_2$, $\phi_3$, $\phi_4$) as design hyperparameters. Using Algorithm 1 for ReLU equalization, we solved compound inequalities to determine the depth hyperparameters. We determine depth hyperparameters ($\phi_1$, $\phi_2$, $\phi_3$, $\phi_4$) $\in \{(1,5,5,3);$ $(1,5,7,2); (1,6,6,2); (1,7,5,2)\}$ corresponding to the minimum values enabling ReLU equalization, resulting in a network global depth ($\phi_1{+}\phi_2{+}\phi_3{+}\phi_4$) of 14. We then varied $m \in \{16, 32, 64, 128, 256\}$ to increase parameters and FLOPs per unit of ReLU in BaseCh networks.

The experimental results, shown in Figure 17, compare the ReLU and FLOPs efficiency with BaseCh and StageCh networks. ReLU and FLOPs efficiency of the derived networks were either similar to or worse than the BaseCh networks. For example, HRN[1,5,5,3] exhibits inferior ReLU (FLOPs) efficiency at higher ReLU (FLOPs) counts compared with BaseCh networks. This underscores the significance of ReLU equalization through width adjustment by altering $\alpha$, $\beta$, and $\gamma$, and demonstrates that *ReLU equalization alone does not yield the desired benefits in HybReNets*.

# E Capacity-Criticality-Tradeoff

## E.1 Investigating Capacity-Criticality-Tradeoff in HybReNets

We conducted additional experiments on various HybReNets to further investigate the Capacity-Criticality Tradeoff phenomenon observed in Figure 6. We progressively reduced the $\alpha$ values in all the HRNs, increasing the proportion of the network's ReLUs in Stage1 (see Table 11). For example, HRN-6x6x2x, HRN-4x6x2x, and HRN-2x6x2x have Stage1 ReLU fractions of 20.4%, 27.8%, and 43.5%, respectively.

We employed both the coarse-grained (DeepReDuce) and fine-grained (SNL) ReLU optimization methods on all the HRNs variants, and the results are shown in Figure 18. Consistent with trends in Figure 6, the wider versions of all four HRNs outperform at higher ReLU counts, while HRNs with $\alpha{=}2$, having a higher proportion of Stage1 ReLUs, excel at lower ReLU counts. For instance, HRN-6x6x2x and HRN-4x6x2x outperform HRN-2x6x2x at higher ReLU counts, whereas HRN-2x6x2x excels at lower ReLU counts.

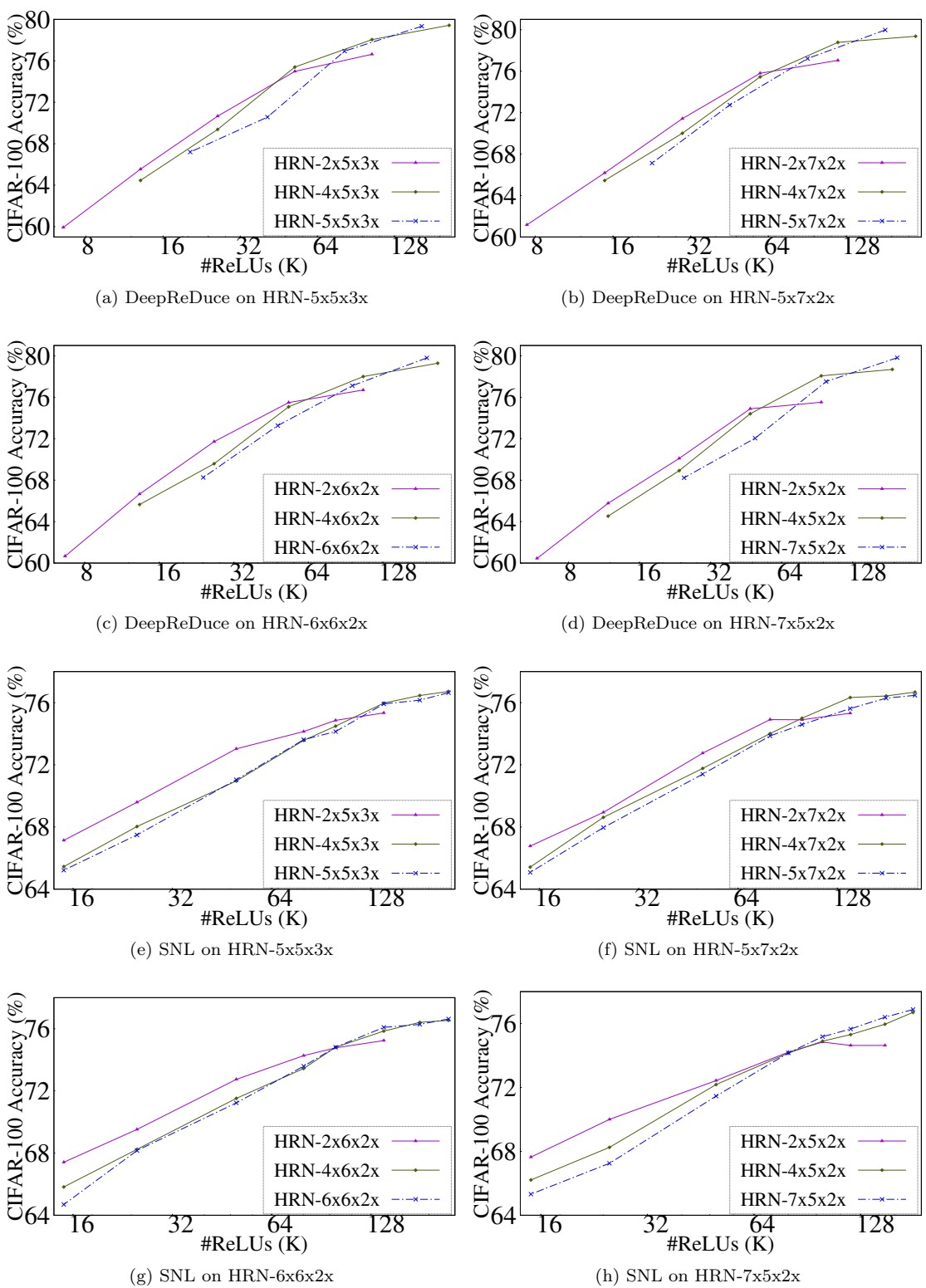

Figure 18: Capacity-Criticality Tradeoff in HRN networks for coarse/fine-grained ReLU optimization DeepReDuce/SNL. HRN networks with lower $\alpha$ posses higher proportion of Stage1 (the least-critical) network's ReLUs, and exhibit superior performance at lower ReLU counts.

## E.2 Intuitive Explanation for Capacity-Criticality Tradeoff

In classical CNNs such as ResNet18 and WRN22x8, networks with a higher proportion of least-critical (Stage1) ReLUs tend to have lower overall ReLU counts. For example, WRN22x8 and ResNet18, used in SNL (Cho et al., 2022b) and SENet (Kundu et al., 2023), have total ReLU counts of 1392.6K and 557K, respectively, with 58.8% and 48.2% of the network's ReLUs in Stage1. This trend is also consistent in HybReNets: HRN-6x6x2x, HRN-4x6x2x, and HRN-2x6x2x have ReLU counts of 401.4K, 294.9K, and 188.4K, with 43.5%, 27.8%, and 20.4% of ReLUs in Stage1, respectively. This raises the question: *what drives better performance at lower ReLU counts*—the proportion of Stage1 ReLUs or the network's total ReLU count?

To investigate this, we compared the performance of networks with the opposite trend: networks with higher ReLU counts and a higher proportion of Stage1 ReLUs. Specifically, we selected ResNet34 and a ReLU efficient variant of ResNet18, ResNet18($m = 16$)-4x4x4x, having a uniform ReLU distribution and used in prior PI-specific network optimization methods Ghodsi et al. (2020); Cho et al. (2022a). The results in Table 12 show that despite having 3.5× fewer ReLUs, compared to ResNet34, ResNet18($m = 16$)-4x4x4x starts performing worse below 50K ReLU counts. This is due to a lower percentage (29.41%) of Stage1 ReLUs in ResNet18($m = 16$)-4x4x4x compared to ResNet34 (47.46%).

| Network | #ReLUs | Stage1(%) | 180K | 100K | 50K | 15K |
|---|---|---|---|---|---|---|
| ResNet34 | 966.7K | 47.46 | 76.35% | 74.55% | 72.07% | 66.46% |
| 4x4x4x | 278.5K | 29.41 | 77.08% | 75.03% | 71.38% | 64.77% |

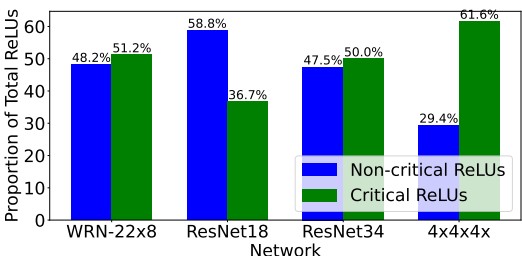

Table 12: Performance comparison (using SNL ReLU optimization) between ResNet34 and a ResNet18 variant (ResNet18($m$=16)-4x4x4x) having a uniform distribution of ReLUs. Stage1(%) indicates the fraction of the network's ReLUs in Stage1. Despite having 3.5× fewer ReLUs, the performance of ResNet18($m$=16)-4x4x4x remains inferior to ResNet34 when the ReLU count falls below 50K. This suggests that the *key determinant* for superior performance at very low ReLU counts is the fraction of least-critical ReLUs, rather than the network's total ReLU count.

Figure 19: Networks with a higher fraction of least-critical (i.e., Stage1) ReLUs, such as ResNet18 and ResNet34, drop a smaller fraction of their critical ReLUs. In contrast, WRN22x8 and 4x4x4x drop a larger fraction of their critical ReLUs to achieve a ReLU count of 25K. This occurs *regardless* of their initial ReLU counts in baseline network.

To understand why a higher fraction of Stage1 ReLUs leads to better performance at lower ReLU counts, we examined the ReLU dropping strategies used in prior ReLU optimization techniques (Kundu et al., 2023; Cho et al., 2022b; Jha et al., 2021). They consistently showed that Stage1 ReLUs are the least critical and are dropped first to achieve very low ReLU counts. Consequently, networks with a higher proportion of least-critical ReLUs drop a smaller fraction of their critical ReLUs when aiming for very low ReLU counts. This is illustrated in Figure 19, where WRN22x8 and ResNet18($m = 16$)-4x4x4x drop a higher fraction of their critical ReLUs compared to ResNet18 and ResNet34 to reach a ReLU count of 25K. Dropping a higher fraction of critical ReLUs leads to significant accuracy loss and inferior performance. This explains why prior work (Kundu et al., 2023; Cho et al., 2022b) used ResNet18 as a backbone network for ReLU optimization at lower ReLU counts and WRN22x8 at higher ReLU counts.

This observation aligns with findings from Yosinski et al. (2014), which showed that neurons in the middle layers of a network (Stage2 and Stage3) exhibit fragile co-adaptation, making them difficult to relearn. Therefore, dropping more ReLUs from these stages in networks with fewer least—critical (Stage1) ReLUs disrupts the fragile co-adaptation and significantly reduces performance.

## F Fine-grained ReLU Optimization on HybReNet Networks

We employed SNL ReLU optimization (Cho et al., 2022b) on HybReNets to examine the efficacy of fine-grained ReLU optimization on these networks. The results, presented in Figure 20, show that HRNs with SNL ReLU optimization are inferior to the (vanilla) SNL, the ReLU-Accuracy Pareto points reported in Cho et al.

(2022b) (which used WRN22x8 (for #ReLUs > 100K) and ResNet18 (for #ReLUs ≤ 100K) as backbone networks). To further investigate, we first performed ReLU-Thinning, where ReLUs from alternate layers are removed, and then employed SNL ReLU optimization. Employing SNL on ReLU-Thinned HRNs results in an *accuracy boost* of up to **3%** (on CIFAR-100) across all ReLU counts, achieving performance comparable to vanilla SNL. This underscores the significance of ReLU Thinning even for fine-grained ReLU optimization.

These findings are consistent with observations made in Figure 7 and Table 5, demonstrating the limitations of SNL when the distribution of ReLUs is changed, the proportion of Stage1 ReLUs decreases and that in other stages increases. This implies that the benefits of fine-grained ReLU optimization are contingent upon the higher proportion of least-critical ReLUs in the network.

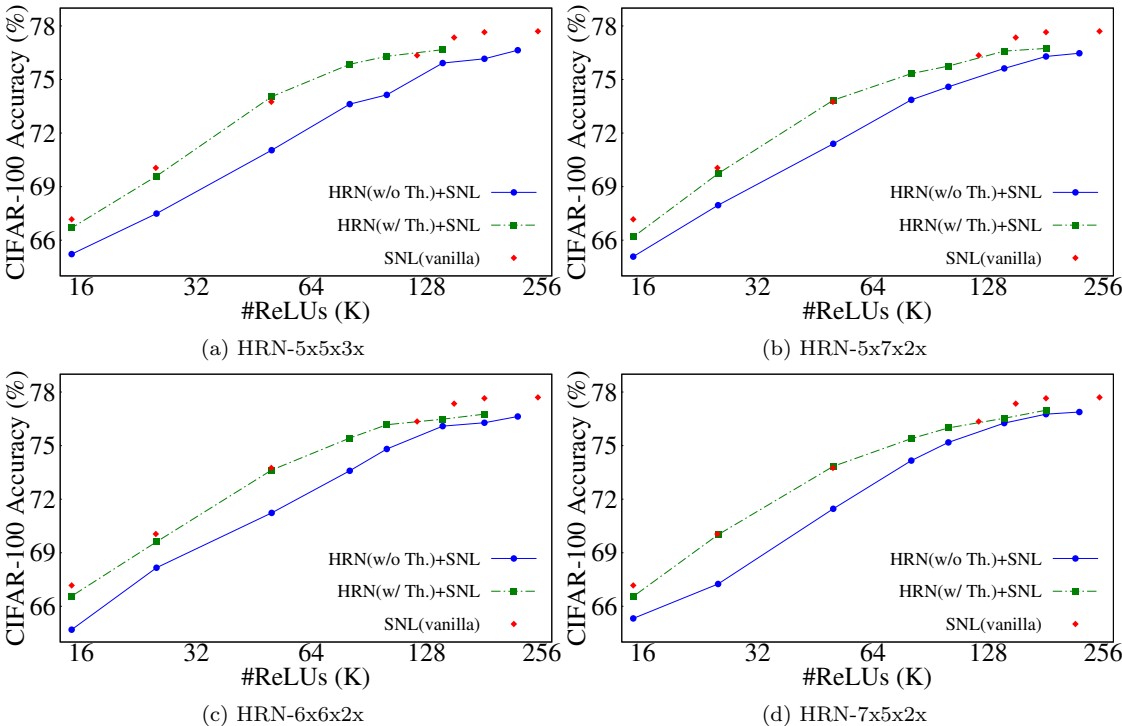

Figure 20: HybReNets with fine-grained ReLU optimization: HRNs using SNL (fine-grained) ReLU optimization exhibit inferior ReLU-Accuracy tradeoff compared to the vanilla SNL, employed on ResNet18 and WRN22x8 (Cho et al., 2022b). Interestingly, using ReLU-Thinning (a coarse-grained optimization step used in DeepReDuce) before SNL optimization yields performance on par with the vanilla SNL. This highlights the significance of ReLU Thinning in ReLU optimization, even in the context of fine-grained ReLU optimization.

To better understand the effectiveness of ReLU Thinning in the context of fine-grained ReLU optimization, we analyzed the total number of ReLUs within a network before and after applying the optimization. For instance, in the HRN-5x7x2x model, the SNL algorithm identified 50K essential ReLUs from an initial pool of 363.5K ReLUs, subsequently eliminating 312.5K ReLUs. However, when we ReLU-Thinning is applied prior to SNL optimization, the SNL algorithm finds 50K critical ReLUs from a reduced pool of 181.25K ReLUs and dropped 131.25K ReLUs. This is because Thinning eliminates half of the network's ReLUs from alternating layers, regardless of their criticality. Thus, ReLU Thinning effectively reduces the search space required to identify critical ReLUs, resulting in an accuracy boost of 2.44% on CIFAR-100.

## G   Detailed Analysis of ReLU-Accuracy Pareto Points

In this section, we describe the ReLU optimization steps used to construct ReLU-Accuracy Pareto frontier shown in Figure 11 (a,b) and Table 7. When employing ReLU-reuse, we fixed $m$=16; however, the network's

FLOPs count reduces due to the group-wise convolution. For instance, HRN-2x6x3x with a ReLU-reuse factor of four has 370.1M FLOPs, while the baseline HRN-2x6x3x ($m$=16) network has 527.4M FLOPs, when used on CIFAR-100 (see Table 14). However, due to specific implementation constraints of group convolution in Microsoft SEAL (SEAL), we calculated the HE latency without considering the FLOP reduction from group convolution in ReLU-reuse.

Table 13: Network configurations and ReLU optimization steps employed for the HybReNet points in Table 7. Accuracies (TinyImageNet) are separately shown for vanilla KD (Hinton et al., 2015) and DKD (Zhao et al., 2022), highlighting the benefits of improved architectural design and distillation method. Re2 denotes ReLU-reuse, and used for efficient PI at very low ReLU counts.

| HybReNet | $m$ | ReLU optimization steps | | | #ReLU | #FLOPs | Accuracy(%) | | Acc./ReLU |
|---|---|---|---|---|---|---|---|---|---|
| | | Culled | Thinned | Re2 | | | KD | DKD | |
| 5x5x3x | 16 | NA | S1+S2+S3+S4 | NA | 653.3K | 4216.4M | 65.76 | 67.58 | 0.10 |
| 2x5x3x | 32 | S1 | S2+S3+S4 | NA | 417.8K | 2841.7M | 64.11 | 66.10 | 0.16 |
| 5x5x3x | 8 | NA | S1+S2+S3+S4 | NA | 326.6K | 1055.4M | 61.92 | 64.92 | 0.20 |
| 2x5x3x | 8 | S1 | S2+S3+S4 | NA | 104.4K | 178.9M | 56.37 | 58.90 | 0.56 |
| 2x5x3x | 16 | S1 | S2+S3+S4 | 4 | 52.2K | 486.3M | 53.13 | 54.46 | 1.04 |

Table 14: Network configurations and ReLU optimization steps employed for the Pareto points in Figure 11 (a,b), along with and the HybReNet points used for the comparison with SOTA PI methods and ConvNeXt-V2 (Woo et al., 2023) as illustrated in Table 14. Re2 denotes ReLU-reuse, a key method in achieving significantly reduced ReLU counts.

| | Nets | m | ReLU optimization steps | | | #ReLU | #FLOPs | Accuracy(%) | | Acc./ReLU |
|---|---|---|---|---|---|---|---|---|---|---|
| | | | Culled | Thinned | Re2 | | | KD | DKD | |
| CIFAR-100 / HybReNet | 4x6x3x | 16 | NA | S1+S2+S3+S4 | NA | 317.4K | 2061.1M | 80.14 | 81.36 | 0.26 |
| | 2x6x3x | 16 | S1 | S2+S3+S4 | NA | 134.1K | 527.4M | 78.80 | 79.56 | 0.59 |
| | 2x6x3x | 8 | S1 | S2+S3+S4 | NA | 67.1K | 132.2M | 74.84 | 76.91 | 1.15 |
| | 2x6x3x | 16 | S1 | S2+S3+S4 | 4 | 33.5K | 370.1M | 70.93 | 73.89 | 2.21 |
| | 2x6x3x | 16 | S1 | S2+S3+S4 | 8 | 16.6K | 394.8M | 68.17 | 69.70 | 4.19 |
| | 2x6x3x | 16 | S1 | S2+S3+S4 | 16 | 8.3K | 413.4M | 62.44 | 64.65 | 7.78 |
| TinyImageNet / HybReNet | 4x6x3x | 16 | NA | S1+S2+S3+S4 | NA | 1269.8K | 8244.2M | 68.90 | 70.29 | 0.06 |
| | 4x6x3x | 12 | NA | S1+S2+S3+S4 | NA | 952K | 4638.8M | 68.16 | 69.15 | 0.07 |
| | 2x6x3x | 16 | S1 | S2+S3+S4 | NA | 536.6K | 2109.4M | 66.29 | 67.48 | 0.13 |
| | 2x6x3x | 12 | S1 | S2+S3+S4 | NA | 402K | 1187.5M | 64.51 | 65.77 | 0.16 |
| | 2x6x3x | 8 | S1 | S2+S3+S4 | NA | 268.3K | 528.6M | 60.97 | 64.02 | 0.24 |
| | 2x6x3x | 16 | S1 | S2+S3+S4 | 4 | 134.1K | 1480.1M | 57.84 | 61.52 | 0.46 |
| | 2x6x3x | 16 | S1 | S2+S3+S4 | 8 | 67.1K | 1579.2M | 54.47 | 56.24 | 0.84 |
| | 2x6x3x | 16 | S1 | S2+S3+S4 | 16 | 33.5K | 1653.5M | 49.13 | 49.96 | 1.49 |
| TinyImageNet / ConvNeXt | T | 96 | S1 | S2+S3+S4 | NA | 1622K | 11801M | 68.32 | 69.85 | 0.04 |
| | N | 80 | S1 | S2+S3+S4 | NA | 1278K | 9080.2M | 66.73 | 68.75 | 0.05 |
| | P | 64 | S1 | S2+S3+S4 | NA | 720.9K | 3435.7M | 65.42 | 67.08 | 0.09 |
| | F | 48 | S1 | S2+S3+S4 | NA | 540.7K | 1935M | 64.23 | 65.72 | 0.12 |
| | A | 40 | S1 | S2+S3+S4 | NA | 450.6K | 1345.1M | 63.23 | 64.08 | 0.14 |

# H  Extended Discussion

## H.1  Constraints for the Simultaneously Optimizing ReLU and FLOPs Efficiency

Classical CNNs often follow the convention of doubling the filter count when downsampling feature maps by a factor of two to prevent representational bottlenecks (Szegedy et al., 2016). This results in a fixed stagewise channel multiplication factor of $\alpha = \beta = \gamma = 2$ for most classical networks. Even in the design

of state-of-the-art FLOPs-efficient vision models, such as RegNet (Radosavovic et al., 2020), the stagewise channel multiplication factors are restricted to $1.5 \leq (\alpha, \beta, \gamma) \leq 3$. Conventionally, network design paradigms prioritize FLOPs efficiency, so the stagewise multiplication factor is typically conservative since the FLOPs count has a quadratic dependence on the channel counts.

To optimize both ReLU and FLOPs efficiency under PI constraints, higher $\alpha$ and $\beta$ values and a lower $\gamma$ value are required (see Figure 3(e,f)). Restricting $\alpha$, $\beta$, and $\gamma$ in classical FLOPs efficient networks (e.g., ResNet and WideResNets) primarily harms the network's ReLU efficiency. Also, the prior approach of designing ReLU-efficient networks by increasing $\alpha$, $\beta$, and $\gamma$ homogeneously (e.g., $\alpha = \beta = \gamma = 4$ in Ghodsi et al. (2020); Cho et al. (2022a)) results in poor FLOPs efficiency due to higher $\gamma$.

## H.2 Achieving ReLU and FLOPs Efficiency in HybReNets by Regulating FLOPs in Deeper Layers

ReLU equalization in a four-stage network imposes specific constraints on $\beta$ and $\gamma$ values, with $\beta\gamma < 16$ and $\gamma < 4$. These constraints effectively limit the explosion of FLOPs in deeper layers, making HRN networks more FLOPs efficient than their StageCh counterparts, which use homogeneous sets of $\alpha$, $\beta$, and $\gamma$. This homogeneity in StageCh networks leads to a rapid increase in FLOPs due to the multiplicative effect of stagewise multiplication factors. Hence, even a slight increase in these factors can significantly increase FLOPs count, especially in deeper layers.

To illustrate this, we computed the normalized FLOPs in ResNet18-based StageCh networks and compared them with HRNs. The normalized FLOPs in Stage3 and Stage4 of ResNet18 are expressed as $\frac{\alpha^2\beta^2}{16}$ and $\frac{\alpha^2\beta^2\gamma^2}{64}$, respectively. Thus, a network with $\gamma$=2 would have equal FLOPs in Stage3 and Stage4. This is evident from the normalized FLOPs ratios in HRN-5x7x2x, HRN-7x5x2x, and HRN-6x6x2x (Table 15).

In particular, the constraints on $\gamma$ (i.e., $\gamma < 4$) keep the ReLU count of Stage4 lower than that of Stage3 (the most critical stage), and restrict the FLOPs in Stage4. While further limiting $\alpha$ and $\beta$ values can reduce the network's FLOPs, this would also reduce the proportion of the most critical (Stage3) ReLUs. Therefore, unlike StageCh networks, a criticality-aware design streamlines both the ReLU and FLOPs in networks by preventing superfluous FLOPs in deeper layers and *maximizes the FLOPs utilization for a given ReLU count*.

Table 15: Stagewise FLOPs and ReLU trend variations with $\alpha$, $\beta$, and $\gamma$ in (ReLU-efficient) StageCh and HRNs. The least (most) critical ReLUs are colored in red (blue). Evidently, ReLU equalization in HRNs restricts the growth of FLOPs in deeper layers, achieving ReLU efficiency comparable to StageCh networks but with fewer FLOPs. *This enables the right balance between ReLU and FLOPs efficiency in HRNs.*

| | Stage1 | Stage2 | Stage3 | Stage4 | | $(\alpha,\beta,\gamma)$=2 | $2<(\alpha,\beta,\gamma)<4$ | $(\alpha,\beta,\gamma)$=4 | $(\alpha,\beta,\gamma)$>4 |
|---|---|---|---|---|---|---|---|---|---|
| FLOPs | 1 | $\frac{\alpha^2}{4}$ | $\frac{\alpha^2\beta^2}{16}$ | $\frac{\alpha^2\beta^2\gamma^2}{64}$ | Layerwise FLOPs | constant | increasing ($\uparrow$) | increasing ($\uparrow\uparrow$) | increasing ($\uparrow\uparrow\uparrow$) |
| ReLUs | 1 | $\frac{\alpha}{4}$ | $\frac{\alpha\beta}{16}$ | $\frac{\alpha\beta\gamma}{64}$ | Layerwise ReLUs | decreasing ($\downarrow\downarrow$) | decreasing ($\downarrow$) | constant | increasing ($\uparrow$) |

| | $(\alpha,\beta,\gamma) = (2, 2, 2)$ | | | | $(\alpha,\beta,\gamma) = (3, 3, 3)$ | | | | $(\alpha,\beta,\gamma) = (4, 4, 4)$ | | | | $(\alpha,\beta,\gamma) = (6, 6, 6)$ | | | |
|---|---|---|---|---|---|---|---|---|---|---|---|---|---|---|---|---|
| | Stage1 | Stage2 | Stage3 | Stage4 | Stage1 | Stage2 | Stage3 | Stage4 | Stage1 | Stage2 | Stage3 | Stage4 | Stage1 | Stage2 | Stage3 | Stage4 |
| FLOPs | 64 | 64 | 64 | 64 | 64 | 144 | 324 | 729 | 64 | 256 | 1024 | 4096 | 64 | 576 | 5184 | 46656 |
| ReLUs | 64 | 32 | 16 | 8 | 64 | 48 | 36 | 27 | 64 | 64 | 64 | 64 | 64 | 96 | 144 | 216 |

| | HRN-5x7x2x | | | | HRN-7x5x2x | | | | HRN-6x6x2x | | | | HRN-5x5x3x | | | |
|---|---|---|---|---|---|---|---|---|---|---|---|---|---|---|---|---|
| | Stage1 | Stage2 | Stage3 | Stage4 | Stage1 | Stage2 | Stage3 | Stage4 | Stage1 | Stage2 | Stage3 | Stage4 | Stage1 | Stage2 | Stage3 | Stage4 |
| FLOPs | 64 | 400 | 4900 | 4900 | 64 | 784 | 4900 | 4900 | 64 | 576 | 5184 | 5184 | 64 | 400 | 2500 | 5625 |
| ReLUs | 64 | 80 | 140 | 70 | 64 | 112 | 140 | 70 | 64 | 96 | 144 | 72 | 64 | 80 | 100 | 75 |

## H.3 Explanation of Accuracy Saturation in StageCh Networks Through Deep Double Descent

A distinct trend in the ReLU-Accuracy tradeoff has been observed (see Figure 3(a,b)) as the width of models increases by augmenting $\alpha$, $\beta$, and $\gamma$. Specifically, accuracy initially increases with increasing values of $\alpha$, $\beta$, and $\gamma$, reaches a saturation point, then increases again at higher ReLU counts. This trend is more prominent in models with smaller depth, such as ResNet18, and disappears in deeper models such as ResNet56, where performance does not improve once accuracy saturated.

The observed saturation trend in the ReLU-Accuracy tradeoff can be explained by the "model-wise deep double descent" phenomenon, which becomes more pronounced with higher label noise (Belkin et al., 2019; Nakkiran et al., 2021; Somepalli et al., 2022). In particular, with label noise, a U-shaped curve appears in the classical (under-parameterized) regime due to a bias-variance tradeoff. Whereas, in the over-parameterized regime, accuracy improves due to the regularization enabled by the strong inductive bias of the network. However, with zero label noise, the test error plateaus around the interpolation threshold, resulting in a flat trend similar to the one shown in Figure 3(a,b) instead of a U-shaped curve.

### H.4 Why RegNet and ConvNeXt Models are Selected for Our Case Study?

RegNets are models designed using a semi-automated network design method, parameterized by the stage compute ratio ($\phi_1$, $\phi_2$, $\phi_3$, $\phi_4$), base channel count ($m$), and stagewise channel multiplication factors within the range $1.5 \leq (\alpha, \beta, \gamma) \leq 3$. In contrast, ConvNeXts are redesigned ResNets with modified values of $m$ and $\phi_1$, $\phi_2$, $\phi_3$, $\phi_4$. For example, the ConvNeXt-T model has a stage compute ratio of [3, 3, 9, 3], different from the [3, 4, 6, 3] ratio used in ResNet34, and a base channel count of $m$=96, higher than the $m$=64 used in ResNet34. These unconstrained design choices in RegNets and the modified depth and width in ConvNeXt models make them suitable for our case study, where we investigate their impact on ReLU distribution and the balance between ReLU and FLOPs efficiency.

The original ConvNeXt model uses the PI-unfriendly GELU activation function Hendrycks & Gimpel (2016) and LayerNorm Ba et al. (2016), which require several approximations in ciphertext evaluation (Fan et al., 2022; Zimerman et al., 2024). Therefore, we adapted ResNet models to incorporate ConvNeXt features, specifically the stagewise channel allocation and the stage compute ratio from ConvNeXt models. For experiments with RegNets, we opted for the RegNet-x variant instead of RegNet-y, as the latter employs a sigmoid activation function in its squeeze-and-excitation blocks.

### H.5 Potential of ReLU Equalization as a Unified Network Design Principle

Deep learning has advanced significantly in recent years with the development of sophisticated neural network architectures. Traditionally, researchers have relied on manual network design techniques, such as ResNet (He et al., 2016), ResNeXt (Xie et al., 2017), and ConvNeXt (Liu et al., 2022). Neural architecture search (NAS) methods (Liu et al., 2018; Tan & Le, 2019; Howard et al., 2019; Tan et al., 2019) have also shown promise.

However, these approaches have their own limitations. Manual techniques can become more complex and lead to suboptimal models as the number of design hyperparameters increases. NAS methods often lack interpretability and may not generalize beyond restricted settings. Moreover, both methods require substantial computational resources to find optimal design hyperparameters when designing networks from scratch.

A semi-automated design technique like RegNets (Radosavovic et al., 2020) offers an interpretable network design and automates the process of finding the optimal population of networks, generalizable across a wide range of settings. However, it still requires training thousands of models to iteratively narrow down the search space, making it expensive when models are designed from scratch.

We analyzed the ReLUs' distribution in SOTA semi-automated models (RegNets) and manually designed models (ConvNeXt) and made the following observations:

- The specific values of width and depth hyperparameters, determined by a quantized linear function in RegNet models, position the networks' ReLUs in their criticality order (see Figure 21(a))

- Similar to HRNs with $\alpha$=2, in ConvNeXt models, the distribution of ReLUs, except for Stage1, follows their criticality order.

In particular, the proportion of the most critical (Stage3) ReLUs increases from 20.3% in ResNet34 to 30.2% in ConvNeXt-T and 34.8% in ConvNeXt-N model while reducing the proportion of less critical ReLUs (see Figure 21(b)). The stagewise channel allocation in ConvNeXt models and HRNs (with $\alpha$=2) shows that both models allocate an identical number of channels in the deeper stages. HRNs allocate fewer channels during the initial stages. These observations demonstrate the generality of ReLU equalization as a design principle, even when designing FLOPs efficient models.

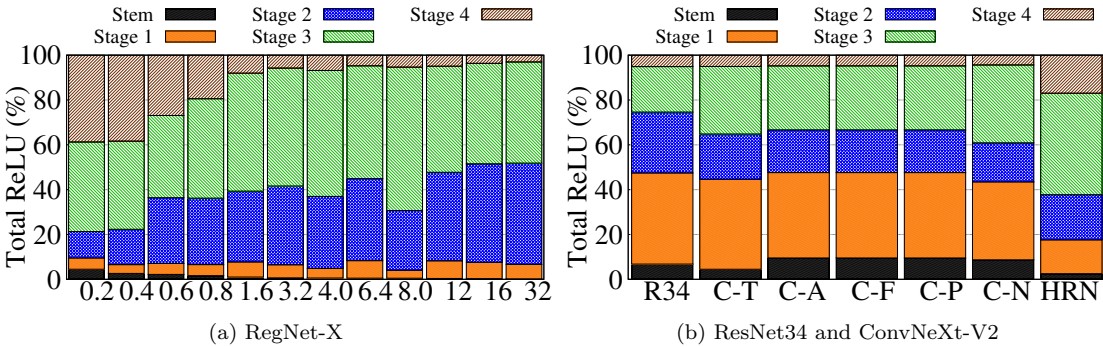

Figure 21: Analysis of ReLUs' distribution in RegNet-X and ConvNeXt-V2 architectures. In all RegNet-X models, ReLUs are arranged precisely according to their criticality order. ConvNeXt models, including their T, A, F, P, and N variations (Woo et al., 2023), have a higher proportion of ReLUs in the most critical stage (Stage3) and fewer in the less critical stage (Stage1) due to modified depth and width design hyper-parameters. In contrast, the distribution of ReLUs in the HRN-4x6x3x model (designed from the ResNet34 network) strictly adheres to their criticality order.

- ConvNext V2-T $m$=96 [96, 192, 384, 768]
- ConvNext V2-N $m$=80 [80, 160, 320, 640]
- ConvNext V2-P $m$=64 [64, 128, 256, 512]
- ConvNext V2-F $m$=48 [48, 96, 192, 384]
- ConvNext V2-A $m$=40 [40, 80, 160, 320]

- HRN-2x6x2x $m$=32 [32, 64, 384, 768]
- HRN-2x5x2x $m$=32 [32, 64, 320, 640]
- HRN-2x5x3x $m$=16 [16, 32, 160, 480]
- HRN-2x6x2x $m$=16 [16, 32, 192, 384]
- HRN-2x5x2x $m$=16 [16, 32, 160, 320]

ReLU equalization, on the other hand, requires only prior knowledge of the stagewise criticality of baseline network, which is often consistent within a specific model family and requires training very few models. Moreover, ReLU equalization can be used to design both FLOPs and ReLU-efficient neural networks. Thus, ReLU equalization offers a new perspective to simplify network design, improve interpretability, and achieve both FLOPs and ReLU efficiency.

# I Performance Comparison of HybReNets vs. Classical Networks for ReLU-reuse

We compared ReLU-reuse against the conventional scaling method (channel/feature-map scaling) used in DeepReduce (Jha et al., 2021) on both classical networks and HRNs. First, we applied ReLU-reuse to all convolutional layers of the networks, reducing the ReLUs by a factor of $N \in \{2, 4, 8, 16\}$. For $N = 2$, we used the naive ReLU reduction method, as it outperformed the proposed ReLU-reuse (see Figure 9).

Experimental results show that for ResNet18 BaseCh networks, ReLU-reuse performs inferior to conventional scaling methods, and this performance gap increases for networks with higher $m$. In contrast, for HRN networks, ReLU-reuse surpasses conventional scaling at higher ReLU reduction factors (i.e., at very low ReLU counts). However, at higher ReLU counts, particularly for a reduction factor of two, the information loss incurred from feature maps division outweighs the benefits of ReLU-reuse, leading to inferior performance. This observation holds even for networks with uniform ReLUs distribution, such as ResNet18($m$=16)-4x4x4x.

Furthermore, we repeat the experiments on ReLU-Thinned networks, as conventional scaling is performed on Thinned networks in Jha et al. (2021). We dropped ReLUs from alternate convolutional layers and applied ReLU-reuse in the remaining layers. The results are shown in Figure 23. Since ReLU-reuse is now only employed in half of the total number of layers, the cumulative information loss caused by the loss of cross-channel information in feature-map divisions is reduced. Consequently, the performance of ReLU-reuse is further improved. This improvement is noticeable from the change in the performance gap between ReLU-reuse and conventional scaling for all the networks, as shown in Figure 23.

To summarize, the effectiveness of ReLU-reuse depends on the network architecture and the ReLU reduction factor. ReLU-reuse is effective for networks with (partial/full) ReLU equalization, in contrast to classical

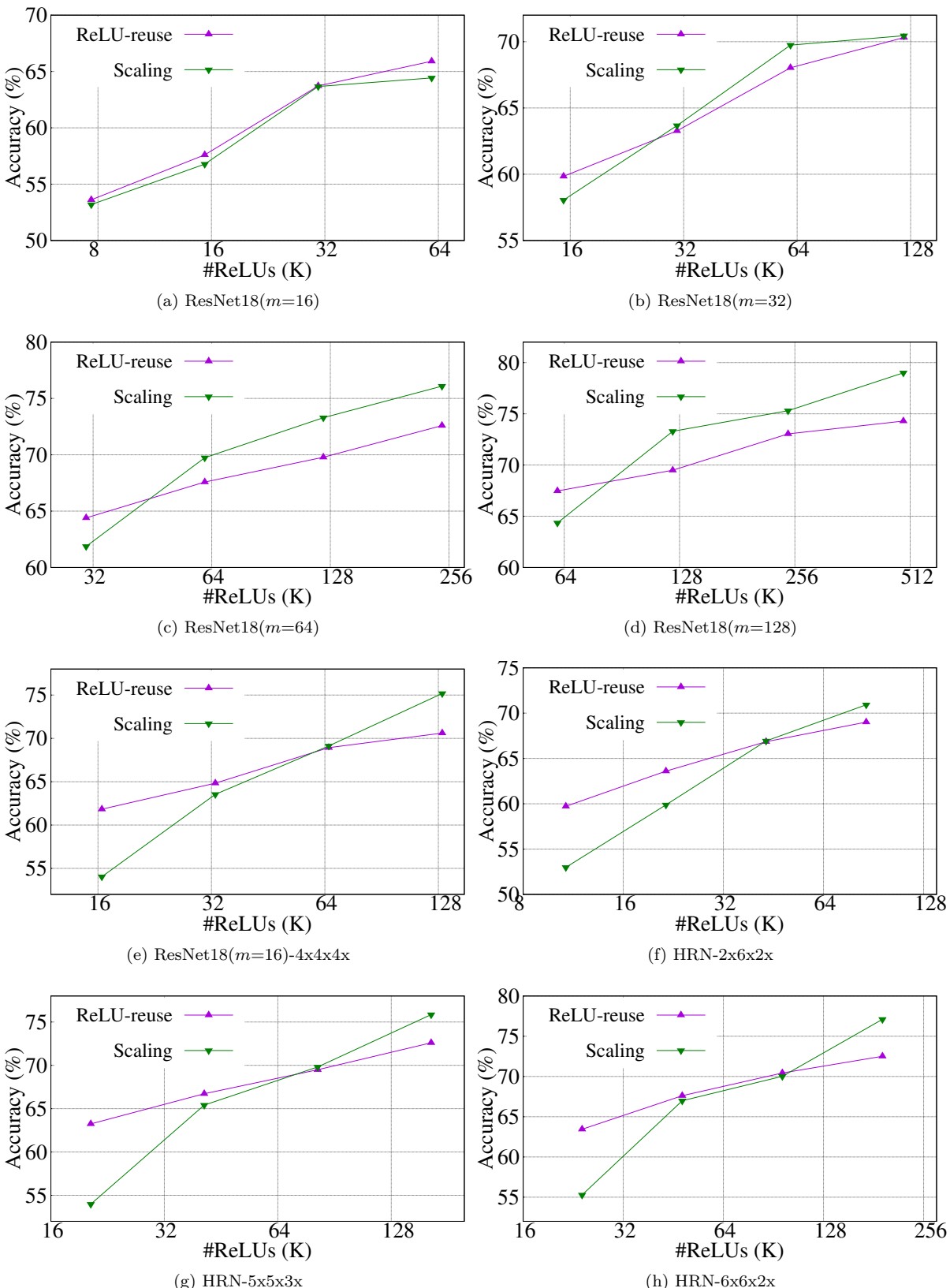

Figure 22: Performance comparison (CIFAR-100) of ReLU-reuse vs conventional scaling (used in coarse-grained ReLU optimization (Jha et al., 2021)) when ReLU-reuse is employed after every convolution layer.

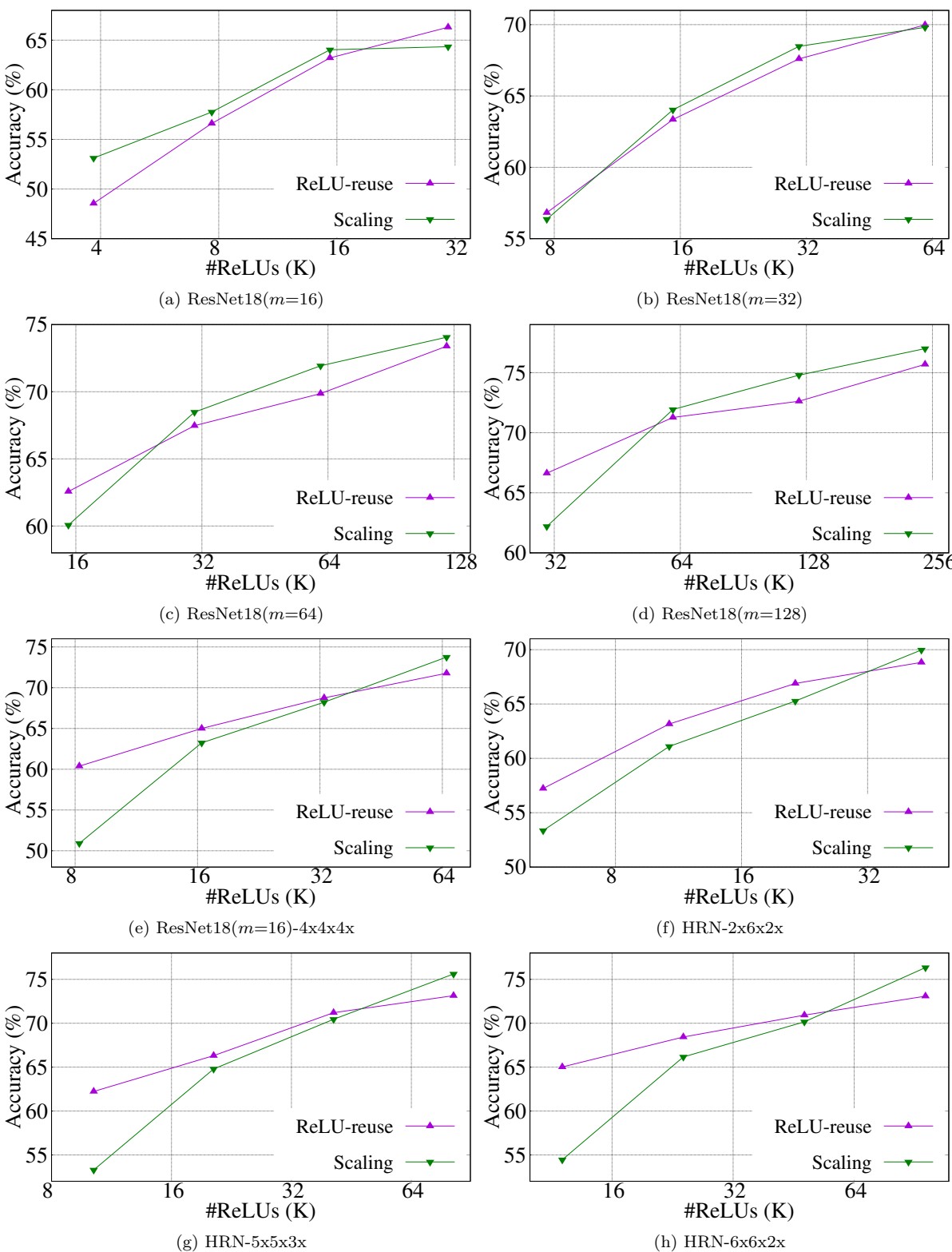

Figure 23: Performance comparison (CIFAR-100) of ReLU-reuse vs conventional scaling (used in coarse-grained ReLU optimization (Jha et al., 2021)) when ReLU-reuse is employed on ReLU-Thinned networks. That is, first ReLU is dropped from every-alternate layers, and then ReLU-reuse is applied in remaining layers (see Algorithm 2).

networks, and scales well with higher ReLU reduction factors. Therefore, ReLU-Thinned HRN networks combined with ReLU-reuse significantly improve performance at very low ReLU counts, which we incorporate in the ReLU optimization pipeline designed for HybReNets (shown in Algorithm 2).

---

**Algorithm 2** ReLU optimization steps employed in HybReNets(HRNs)

---

**Input:** A network $Net$ with $D$ stages $S_1, S_2, ..., S_D$ and $C$, a sorted list of stages from least to most critical
**Output:** ReLU optimized versions of $Net$

1: **if** the least critical stage C[1] dominates the distribution of ReLUs **then**
2:     $S_k = C[1]$                                                                   ▷ Get the least critical stage
3:     $Net = Net - S_k$                                                          ▷ Cull the least critical Stage $S_k$
4: **end if**
5: $Net_i^T = Thin(Net)$                                                      ▷ Thin the remaining stages
6: $Net_i^C = ScaleCh(Net_i^T, \alpha{=}0.5)$                               ▷ Channel scaled by 0.5x
7: $Net_i^{R4} = ReuseReLU\ (Net_i^T, \text{Sc}{=}4)$                    ▷ ReLU-reuse with scaling factor 4
8: $Net_i^{R8} = ReuseReLU\ (Net_i^T, \text{Sc}{=}8)$                    ▷ ReLU-reuse with scaling factor 8
9: $Net_i^{R16} = ReuseReLU\ (Net_i^T, \text{Sc}{=}16)$                  ▷ ReLU-reuse with scaling factor 16
10: $Nets\ += Net, Net_i^T, Net_i^C, Net_i^{R4}, Net_i^{R8}, Net_i^{R16}$     ▷ Apply KD to each Net
11: **return** $Nets$

---

## J Design of Experiments and Training Procedure

**Sweeping the width hyperparameters for ReLU efficiency experiments:** To examine the effect of various network widening methods (primarily BaseCh and StageCh), we conducted experiments with ResNet18 as backbone architecture in Figure 3 (a,b). We reduced the base channel count in ResNet18 to $m{=}16$ (from $m{=}64$) to enable a fair comparison with other ResNet models, such as ResNet20, ResNet32, and ResNet56, which have $m{=}16$. These ResNet models have [16, 32, 64] #channels in their successive stages, while that in (original) ResNet18 is [64, 128, 256, 512]. For BaseCh networks, we sweep $m \in \{16,32,64,128,256\}$ and for StageCh networks, we sweep $(\alpha, \beta, \gamma) = (2, 2, 2)$ to $(8, 8, 8)$, homogeneously.

**Training methodology and datasets:** We performed our experiments on CIFAR-100 (Krizhevsky et al., 2010) and TinyImageNet (Le & Yang, 2015; Yao & Miller, 2015), as prior PI-specific network optimization studies (Jha et al., 2021; Cho et al., 2022b; Kundu et al., 2023) used these datasets. CIFAR-100 consists of 100 classes, each with 500 training images and 100 test images of resolution 32×32. TinyImageNet includes 200 classes, each with 500 training images and 50 validation images with a resolution of 64×64.

For training on CIFAR-100 and TinyImageNet, we used a cosine annealing learning rate scheduler (Loshchilov & Hutter, 2016) with an initial learning rate of 0.1, a mini-batch size of 128, a momentum of 0.9, and a weight decay factor of 5$e$-4. We trained the networks for 200 epochs on both datasets, with an additional 20 warmup epochs for Decoupled KD (Zhao et al., 2022).

For DeepReDuce and KD experiments in Tables 6, Table 13, and Table 14, we employed Hinton's knowledge distillation (Hinton et al., 2015) with a temperature of 4 and a relative weight to cross-entropy loss as 0.9.

For SNL, we train the baseline networks using the aforementioned methodology; however, we used their default implementation for mask generation, fine-tuning, and knowledge distillation. When employing Decoupled knowledge distillation (Tables 6, 13, and 14), we set the relative weight of target class KD as one and vary the weight of non-target class KD as {0.8, 1, 2, 6}. For the KD experiment, we consistently employed ResNet18 as the teacher model for a fair comparison across the studies.

**Runtime measurement:** We adopted the methodology from Garimella et al. (2023) to compute the runtime of a single (private) inference. Specifically, we used Microsoft SEAL to compute the homomorphic encryption (HE) latency for convolution and fully connected operations, and DELPHI (Mishra et al., 2020) to compute the garbled-circuit (GC) latency for ReLU operations. Our experimental setup involved an AMD EPYC 7502 server with 2.5 GHz, 32 cores, and 256 GB RAM. The client and server were simulated as two separate processes operating on the same machine. We set the number of threads to four to compute the GC latency.

Note that for HRNs with ReLU-reuse, we calculated the HE latency without accounting for the FLOP reduction from group convolution in ReLU-reuse due to the implementation constraints in Microsoft SEAL.

## K    Network Architecture of HybReNets

Table 16 shows the architectural comparison between ResNet18, WRN22x8, and four specific HRNs: HRN-5x5x3x, HRN-5x7x2x, HRN-6x6x2x, and HRN-7x5x2x. Compared to ResNet18, the HRNs allocate fewer channels in the initial stages and increase those in the deeper stages of the network. This strikes the right balance between ReLUs and FLOPs efficiency. Among the four HRNs, HRN-5x5x3x offers a slightly better balance between ReLU and FLOPs efficiency.

Table 16: Comparison of WideResNet22x8 and ResNet18 architectures, which are commonly used as baseline networks in PI-specific ReLU optimization techniques (Kundu et al., 2023; Cho et al., 2022b; Jha et al., 2021), with our proposed HybReNets (highlighted in bold). Unlike conventional WideResNets and ResNets, the strategic channel allocation in the subsequent stages of HybReNets streamlines the network's ReLUs and FLOPs, optimizing both ReLU and FLOPs efficiency simultaneously. The last rows compare their FLOPs and ReLU counts, along with baseline accuracy on CIFAR-100.

| Stages | output size | WRN22x8 | ResNet18 | HRN-5x5x3x | HRN-5x7x2x | HRN-6x6x2x | HRN-7x5x2x |
|---|---|---|---|---|---|---|---|
| Stem | $32 \times 32$ | [3×3, 16] | [3×3, 64] | [3×3, 16] | [3×3, 16] | [3×3, 16] | [3×3, 16] |
| Stage1 | 32×32 | $\begin{bmatrix} 3{\times}3,\ 128 \\ 3{\times}3,\ 128 \end{bmatrix}{\times}3$ | $\begin{bmatrix} 3{\times}3,\ 64 \\ 3{\times}3,\ 64 \end{bmatrix}{\times}2$ | $\begin{bmatrix} 3{\times}3,\ 16 \\ 3{\times}3,\ 16 \end{bmatrix}{\times}2$ | $\begin{bmatrix} 3{\times}3,\ 16 \\ 3{\times}3,\ 16 \end{bmatrix}{\times}2$ | $\begin{bmatrix} 3{\times}3,\ 16 \\ 3{\times}3,\ 16 \end{bmatrix}{\times}2$ | $\begin{bmatrix} 3{\times}3,\ 16 \\ 3{\times}3,\ 16 \end{bmatrix}{\times}2$ |
| Stage2 | 16×16 | $\begin{bmatrix} 3{\times}3,\ 256 \\ 3{\times}3,\ 256 \end{bmatrix}{\times}3$ | $\begin{bmatrix} 3{\times}3,\ 128 \\ 3{\times}3,\ 128 \end{bmatrix}{\times}2$ | $\begin{bmatrix} 3{\times}3,\ 80 \\ 3{\times}3,\ 80 \end{bmatrix}{\times}2$ | $\begin{bmatrix} 3{\times}3,\ 80 \\ 3{\times}3,\ 80 \end{bmatrix}{\times}2$ | $\begin{bmatrix} 3{\times}3,\ 96 \\ 3{\times}3,\ 96 \end{bmatrix}{\times}2$ | $\begin{bmatrix} 3{\times}3,\ 112 \\ 3{\times}3,\ 112 \end{bmatrix}{\times}2$ |
| Stage3 | 8×8 | $\begin{bmatrix} 3{\times}3,\ 512 \\ 3{\times}3,\ 512 \end{bmatrix}{\times}3$ | $\begin{bmatrix} 3{\times}3,\ 256 \\ 3{\times}3,\ 256 \end{bmatrix}{\times}2$ | $\begin{bmatrix} 3{\times}3,\ 400 \\ 3{\times}3,\ 400 \end{bmatrix}{\times}2$ | $\begin{bmatrix} 3{\times}3,\ 560 \\ 3{\times}3,\ 560 \end{bmatrix}{\times}2$ | $\begin{bmatrix} 3{\times}3,\ 576 \\ 3{\times}3,\ 576 \end{bmatrix}{\times}2$ | $\begin{bmatrix} 3{\times}3,\ 560 \\ 3{\times}3,\ 560 \end{bmatrix}{\times}2$ |
| Stage4 | 4×4 | | $\begin{bmatrix} 3{\times}3,\ 512 \\ 3{\times}3,\ 512 \end{bmatrix}{\times}2$ | $\begin{bmatrix} 3{\times}3,\ 1200 \\ 3{\times}3,\ 1200 \end{bmatrix}{\times}2$ | $\begin{bmatrix} 3{\times}3,\ 1120 \\ 3{\times}3,\ 1120 \end{bmatrix}{\times}2$ | $\begin{bmatrix} 3{\times}3,\ 1152 \\ 3{\times}3,\ 1152 \end{bmatrix}{\times}2$ | $\begin{bmatrix} 3{\times}3,\ 1120 \\ 3{\times}3,\ 1120 \end{bmatrix}{\times}2$ |
| FC | $1 \times 1$ | [8 × 8, 512] | [4 × 4, 512] | [4 × 4, 1200] | [4 × 4, 1120] | [4 × 4, 1152] | [4 × 4, 1120] |
| #FLOPs | | 2461M | 559M | 1055M | 1273M | 1368M | 1328M |
| #ReLUs | | 1393K | 557K | 343K | 379K | 401K | 412K |
| Accuracy | | 81.27% | 79.01% | 78.40% | 78.28% | 78.52% | 78.81% |

Table 17 details the architectural specifics of HRNs with $\alpha{=}2$, which are crucial for performing PI at lower ReLU counts. These HRNs developed using ResNet18 as backbone architecture, leading to stage compute ratios of [2, 2, 2, 2]. Similarly, HRNs designed using ResNet34 as backbone architecture have stage compute ratios [3, 4, 6, 3], same as ResNet34 (For instance, HRN-4x6x3x in Figure 21(b)).

Table 17: Baseline HybReNets, developed using ResNet18 as the backbone architecture, aim for efficient PI with lower ReLU counts. The final rows compare their FLOPs, ReLU counts, and baseline accuracy on the CIFAR-100 dataset.

| Stages | output size | HRN-2x5x3x | HRN-2x7x2x | HRN-2x6x2x | HRN-2x5x2x |
|---|---|---|---|---|---|
| Stem | $32 \times 32$ | [3×3, 16] | [3×3, 16] | [3×3, 16] | [3×3, 16] |
| Stage1 | 32×32 | $\begin{bmatrix} 3\times3,\ 16 \\ 3\times3,\ 16 \end{bmatrix}\times2$ | $\begin{bmatrix} 3\times3,\ 16 \\ 3\times3,\ 16 \end{bmatrix}\times2$ | $\begin{bmatrix} 3\times3,\ 16 \\ 3\times3,\ 16 \end{bmatrix}\times2$ | $\begin{bmatrix} 3\times3,\ 16 \\ 3\times3,\ 16 \end{bmatrix}\times2$ |
| Stage2 | 16×16 | $\begin{bmatrix} 3\times3,\ 32 \\ 3\times3,\ 32 \end{bmatrix}\times2$ | $\begin{bmatrix} 3\times3,\ 32 \\ 3\times3,\ 32 \end{bmatrix}\times2$ | $\begin{bmatrix} 3\times3,\ 32 \\ 3\times3,\ 32 \end{bmatrix}\times2$ | $\begin{bmatrix} 3\times3,\ 32 \\ 3\times3,\ 32 \end{bmatrix}\times2$ |
| Stage3 | 8×8 | $\begin{bmatrix} 3\times3,\ 160 \\ 3\times3,\ 160 \end{bmatrix}\times2$ | $\begin{bmatrix} 3\times3,\ 224 \\ 3\times3,\ 224 \end{bmatrix}\times2$ | $\begin{bmatrix} 3\times3,\ 192 \\ 3\times3,\ 192 \end{bmatrix}\times2$ | $\begin{bmatrix} 3\times3,\ 160 \\ 3\times3,\ 160 \end{bmatrix}\times2$ |
| Stage4 | 4×4 | $\begin{bmatrix} 3\times3,\ 480 \\ 3\times3,\ 480 \end{bmatrix}\times2$ | $\begin{bmatrix} 3\times3,\ 448 \\ 3\times3,\ 448 \end{bmatrix}\times2$ | $\begin{bmatrix} 3\times3,\ 384 \\ 3\times3,\ 384 \end{bmatrix}\times2$ | $\begin{bmatrix} 3\times3,\ 320 \\ 3\times3,\ 320 \end{bmatrix}\times2$ |
| FC | $1 \times 1$ | $[8 \times 8,\ 480]$ | $[4 \times 4,\ 448]$ | $[4 \times 4,\ 384]$ | $[4 \times 4,\ 320]$ |
| #FLOPs | | 179M | 213M | 163M | 119M |
| #ReLUs | | 186K | 201K | 188K | 176K |
| Accuracy | | 75.34% | 75.73% | 75.70% | 75.03% |

