# OpenReview forum: "DeepReShape: Redesigning  Neural Networks for Efficient Private Inference"
_TMLR — Accepted by TMLR_

### Review · Reviewer_T1uq · 2023-12-05

**Summary Of Contributions:**

This paper introduces DeepReShape, an innovative network redesign technique that customizes architectures to meet PI constraints while optimizing for both ReLUs and FLOPs simultaneously. The key insight involves strategically allocating channels to align the network's ReLUs in their criticality order, thereby enhancing both ReLU and FLOP efficiency. DeepReShape streamlines network development through an efficient process, generating networks termed HybReNets. Evaluation on standard PI benchmarks showcases a 2.1% accuracy gain with a 5.2× runtime improvement at iso-ReLU on CIFAR-100 and an 8.7× runtime improvement at iso-accuracy on TinyImageNet.

**Audience:**

Yes

**Claims And Evidence:**

Yes

**Requested Changes:**

See above weaknesses.

**Strengths And Weaknesses:**

**Strengths**

(1)    Extensive Experiments： This paper is filled with rich experiments and observations based on these experiments.

(2)    Important Topic: The paper focuses on an important research area, efficient private inference.

(3)    Well-motivated. This paper offers a comprehensive introduction and analysis of existing channel scaling and ReLU optimization methods.

**Weaknesses**

(1) The originality of this paper is constrained, with its primary contributions centered around heterogeneous channel scaling and hybrid ReLU optimization. However, it is important to note that heterogeneous channel scaling essentially involves adapting homogeneous channel scaling by introducing variations in alpha, beta, and gamma. The hybrid ReLU optimization is also just a combination of existing methods such as ReLU-Thinning and SNL optimization.

(2) Additional experiments are required to strengthen the findings of this paper. While the current work utilizes DELPHI (2020) as the foundational 2PC private inference framework, it's noteworthy that Cheetah (2022) is generally acknowledged for its enhanced efficiency in both linear and nonlinear computations. Cheetah demonstrates a speed advantage of 12 to 30 times faster on end-to-end evaluations compared to DELPHI. Although the primary focus of this paper is on network design, it would be beneficial to demonstrate the compatibility of the proposed techniques with frameworks like Cheetah.

(3) Algorithm 1, responsible for ReLU Equalization and the formation of HybReNet, exhibits limitations in delivering a holistic solution. Its primary output is constraints generated by allocating more ReLUs to critical stages. However, the current approach leaves the user to manually address these constraints, introducing a reliance on trial and error to optimize network design. This aspect raises concerns about the algorithm's user-friendliness and efficiency in providing an automated and robust solution for network optimization.

(4) The authors claim that their proposed algorithms, specifically ReLU equalization and ReLU optimization, are asserted to have a constant time complexity of O(1), regardless of the number of stages N. However, to substantiate this claim convincingly, more in-depth analysis or additional data is needed.

Moreover, the authors' assertion that the proposed approach requires no additional network training lacks clarity. This uncertainty arises because the changes in hyperparameters governing the network width imply a modification to the network structure. Consequently, it seems necessary to fine-tune this altered structure, casting doubt on the claim that additional training is not needed.


(5)  Regarding the second technique, 'ReLU-reuse,' how it alleviates accuracy loss is not explicitly clarified, as ReLU continues to operate solely on the last group (X/N). Moreover, the paper lacks a clear explanation of why feature reuse holds significance within this particular context.

(6)  Typos: On page 5,  ReLU efficiency (see 3(c)--> 3(e)). FLOP efficiency (see 3(d)-->3(f)).

---

> ### Author Response · Authors · 2024-02-05
> **Rebuttal to the Reviwer's comments**
>
> ```1Q: The originality of this paper is constrained, with its primary contributions centered around heterogeneous channel scaling and hybrid ReLU optimization.  ```
>
> *Response:* The need for heterogeneous channel scaling, especially in the context of private inference for simultaneously optimizing ReLUs and FLOP efficiency, has not been previously emphasized. In particular, how each stagewise channel multiplication factor ($\alpha$, $\beta$, and $\gamma$) influences the network’s ReLU and FLOPs efficiency has not been understood. Our sensitivity analysis (shown in Figures 5(e) and 5(f)) sheds light on how each of these factors distinctly influences ReLU and FLOPs efficiency. We observed that for superior ReLU efficiency, higher values of $\alpha$ and $\beta$ are desirable, whereas a lower $\gamma$ value is preferable for FLOPs efficiency. However, prior approaches to channel scaling either uniformly constrained the values of $\alpha$, $\beta$, and $\gamma$ for the sake of FLOPs efficiency (as in WideResNet, $\alpha$=$\beta$=$\gamma$=2), which compromised ReLU efficiency, or uniformly increased them (as seen in CryptoNAS and Sphynx, $\alpha$=$\beta$=$\gamma$=4) to boost ReLU efficiency, but at the cost of FLOPs efficiency.
>
> In this work, we identified the need for heterogeneous channel scaling and introduced a novel design principle (ReLU equalization) to implement it in a compute-efficient manner. This principle was then empirically validated through sensitivity analysis, demonstrating its practical applicability and effectiveness.
>
>
> ```2Q: Runtime measurement with DELPHI vs Cheetah as Cheetah demonstrates a speed advantage of 12 to 30 times faster on end-to-end evaluations compared to DELPHI.```
>
>
> *Response:*  While Cheetah employs rotation-free homomorphic encryption for the faster execution of linear layers, it uses a distinct protocol, Oblivious Transfer (OT), for nonlinear layers. In contrast, the existing benchmarks (DeepReDuce, SNL, SENets, etc.) predominantly utilize Garbled Circuits for nonlinear operations. Therefore, we use Garbled Circuits for nonlinear layers to maintain consistency with the established benchmarks and facilitate a fair and direct comparison with the current SOTA (SENets).
>
> It is important to note that different sets of protocols for private inference significantly change the cost dynamics (in terms of communication, storage, and latency) for linear and nonlinear layers, affecting network optimization goals. For example, CoPriv at NeurIPS’23 [1], which uses OT for nonlinear operations, has different (network) optimization goals. Our proposed method is designed to align with studies predominantly using Garbled Circuits; therefore, it did not adopt the Cheetah framework. Moreover, our primary aim of reducing  FLOPs without sacrificing ReLU efficiency would also benefit systems like Cheetah using rotation-free homomorphic encryption, as it improves the efficiency of linear layer computation without exacerbating the cost of nonlinear operations.
>
>
> ```3Q: The proposed manual design has the constraint of relying on trial and error to optimize network design. This aspect raises concerns about the algorithm's user-friendliness and efficiency in providing an automated and robust solution for network optimization.```
>
> *Response:* In response to concerns about the reliance on trial and error in our design process, we would like to clarify our methodological approach. We have demonstrated that the minimum required values of $\alpha$, $\beta$, and $\gamma$ to meet ReLU equalization conditions are key to the efficiency of our method. Notably, increasing these hyper-parameters beyond their minimum does not necessarily improve the performance (see Figure 15), and we offered an explanation by examining the ReLU distribution within the network as these hyper-parameters are varied (Figure 4(a) and Figure 14). Furthermore, our sensitivity analysis has shown (in Figures 5(e) and 5(f)) that for superior ReLU efficiency, higher $\alpha$ and higher $\beta$ values are needed, a feature that is present in all four of our HybReNet (HRN) models (Table 16). Simultaneously, maintaining lower $\gamma$ values is essential for FLOPs efficiency, which is also a  characteristic in all HRNs. Thus, the balance of $\alpha$, $\beta$, and $\gamma$ values in our HRNs demonstrates the effectiveness of our network design, achieving a well-balanced optimization of ReLU and FLOPs efficiency while minimizing the need for extensive trial and error.

---

> ### Author Response · Authors · 2024-02-05
> **(Continued) Rebuttal to the Reviwers Comments**
>
> ```4Q:  The argument for the constant time complexity of ReLU equalization and employed ReLU optimization required more in-depth analysis or additional data. ```
>
> *Response:*  The constant time complexity for designing HybReNets arises from our method's approach to optimizing a given baseline network (shown in Figure 10)  by solving a compound inequality equation, which provides bounds for $\alpha$, $\beta$, and $\gamma$. To narrow down this search space, we select the minimum values of these hyper-parameters that satisfy the ReLU equalization conditions and justify them by analyzing their ReLUs’ distribution (Figure 4(a) and Figure 14) and providing empirical evidence (Figure 15).
>
> In response to the ```“proposed approach requires no additional network training,”``` we would like to clarify that this is in the context of the approach to narrow down the search space. Specifically, automated design methods such as neural architecture search or even semi-automated design methods (as employed in RegNets) require training thousands of network instances to find the optimal design,  as their search space is huge. Our method, instead, leverages the existing designs and further optimizes them under PI constraints rather than designing them from scratch. Nonetheless, *our method does not claim to obtain optimal design, and a more sophisticated one can find an optimal design*.
>
> We further analyzed the ReLU distribution in state-of-the-art FLOPs-efficient networks,  whether semi-automated or manually designed (e.g., RegNets and ConvNeXt). We found that their ReLU distribution follows that of the HybReNets. Thus, we believe that our approach can further improve the understanding and rationale for existing network design. A more in-depth analysis is presented in Appendix E.5.
>
> ``In response to the ReLU pruning and their constant time complexity in HybReNets,`` we would like to clarify that we intentionally avoid SNL optimization, given its limited effectiveness in networks with atypical ReLU distributions. Instead, we employed coarse-grained ReLU optimization steps (such as ReLU-Thinning) due to its adaptability to diverse ReLU distributions. Notably, coarse-grained ReLU optimization in conventional networks like ResNets requires D-1 (D is the number of stages in a network) iterations, as ReLUs are not arranged in their criticality order, leading to suboptimal results if the ReLU pruning is performed just for a single iteration. However, our HybReNets require only one iteration of ReLU pruning, thanks to the pre-arranged ReLUs by their criticality order. In essence, an improved baseline design can substantially lower the computational complexity of ReLU pruning., from BigO(D) to Big(1).
>
>
> ```5Q: How does ReLU-reuse alleviate accuracy loss as ReLU continues to operate solely on the last group (X/N), and why does feature reuse hold significance within this particular context?  ```
>
> *Response:* We have included an ablation study at the end of our experimental result section (Section 5) to illustrate the significance of feature reuse and the constant number of feature-map divisions in ReLU-reuse. The results are shown in Table 9 (also shown below).
>
> |ReLU reduction factors|#ReLUs |N division| |Proposed|
> |:----|:----|:----|:----|:----|
> | | |w/o reuse|w/ Reuse|3 division|
> |2x reduction (N=2)|434.18K|77.61%|78.19%|77.83%|
> |4x reduction (N=4)|372.74K|75.84%|76.87%|77.60%|
> |8x reduction (N=8)|342.02K|75.43%|75.66%|76.93%|
> |16x reduction (N=16)|326.66K|75.33%|75.47%|76.38%|
>
>
> Specifically, we have shown that by incorporating feature-reuse (and shortcut connections) between the feature map subgroups without ReLU activations, the accuracy of the network increases. This is due to the improved effective receptive field of the neurons in the last group (with ReLU activations), as ReLUs in the last group of feature maps can see a larger subset of feature maps using the skip connections [2]. However, it gives diminishing results when feature maps are divided into more groups (e.g., 8 and 6), likely due to the significant cross-channel information loss. Thus, to alleviate this, we propose a method to have a constant number of feature maps irrespective of the ReLU reduction factor, which provides scalability for optimizing the network for very low ReLU counts.
>
> [1] Zeng et al., CoPriv: Network/Protocol Co-Optimization for Communication-Efficient Private Inference, NeurIPS 2023.
>
> [2] Gao et al., Res2net: A new multi-scale backbone architecture, IEEE TPAMI 2021

---

### Review · Reviewer_7qPP · 2024-01-01

**Summary Of Contributions:**

This paper proposes DeepReShape, which optimizes both ReLUs and FLOPs for private inference (PI). The main novel technique is ReLU equalization, where they scale the channels to align the ReLU counts in the criticality order. Based on this design principle, they design a family of networks called HybReNet. They compare HybReNet to SOTA models in PI and FLOP-efficient models, and show that HybReNet achieves more substantial FLOP reduction with fewer ReLUs, while achieving additional accuracy gain.

**Audience:**

Yes

**Broader Impact Concerns:**

No concern.

**Claims And Evidence:**

Yes

**Requested Changes:**

See the weaknesses above.

**Strengths And Weaknesses:**

Strengths:

1. The ReLU equalization design is novel.

2. The evaluation is comprehensive, and demonstrates the empirical effectiveness of the approach.

Weaknesses:

The main weakness of this paper is that the paper writing is generally confusing. In particular, Section 3 includes a lot of experimental results, but sometimes the connection to the key design principle (i.e., ReLU equalization) is unclear, e.g., regarding the discussion of different input networks and fine-grained VS coarse-grained ReLU optimization.

---

> ### Author Response · Authors · 2024-02-05
> **Rebuttal to the Reviwer's comment**
>
> We thank the reviewer for the positive comments. In response to the reviewer’s feedback on the overall flow of Section 3, we have added a paragraph at the beginning of each subsection to provide context, underscore the motivation behind the experimental studies, and elucidate how these studies are pertinent to our proposed methods and optimization strategies.
>
>
>
> ```1. Regarding the reviewer's query on the connection between our proposed methods and the discussion on selecting various input networks,``` this aspect of our research is focused on identifying crucial network attributes for superior PI performance when using ReLU pruning methods. This investigation has been instrumental in guiding the design of our baseline HybReNets, particularly for targeting low ReLU counts that significantly differ from those at higher ReLU counts.
>
>
>
> ```2. In response to the connection between ReLU equalization and our examination of “fine-grained vs coarse-grained” ReLU optimization,``` this study aimed to examine the effectiveness of fine-grained ReLU pruning beyond the networks with atypical/non-standard  ReLU distribution. Note that the ReLUs’ distribution in our proposed HybReNets significantly differs from the conventional network (e.g., ResNets and WideResNets), and we found that the fine-grained ReLU optimization on HybReNets led to sub-optimal results (see Figure 19). Thus, we employ the coarse-grained ReLU optimization steps. We have provided the justification for the same in Appendix C.4.

---

### Review · Reviewer_CFqf · 2024-01-07

**Summary Of Contributions:**

This paper provides a comprehensive investigation of efficient private inference (PI) in the context of convolutional neural networks (CNN). To this end, the paper first delves into existing approaches proposed for efficient PI and argues that ReLU optimization-based methods neglect the importance of FLOP efficiency in their proposed algorithms. Among the observations made, the authors demonstrate that the distribution of ReLUs in different stages of a CNN can have a considerable impact on FLOP efficiency. Based on this key observation, the paper proposes a new method called "ReLU Equalization" to ensure that the distribution of ReLUs in different stages of the CNN follow their criticality order. Additionally, a new "ReLU-reuse" algorithm is proposed to further enhance the PI efficiency at low rate ReLUs. The experimental results on CIFAR-100 and TinyImageNet datasets demonstrate that the proposed CNNs, coined HybReNets, improve the ReLU and FLOP efficiency of PI models while maintaining a comparable accuracy to existing methods.

**Audience:**

Yes

**Claims And Evidence:**

Yes

**Requested Changes:**

As mentioned above, I believe that despite the contributions of the paper, the writing should be greatly improved. To this end, I believe that the authors should incorporate the feedback in the weaknesses section above. In terms of the importance of the contributions, I will wait to read the review of my peers before making up my mind.

**Strengths And Weaknesses:**

### Strengths:

- The paper provides a comprehensive experimental study on state-of-the-art efficient PI methods and pinpoints the existing fallacies. Researchers working in this area can greatly benefit from the benchmarking of this paper.

- The proposed method is a straightforward implementation of the observations made by the paper.

- Experimental results demonstrate that the proposed method significantly enhances the existing efficient PI methods.

### Weaknesses:

- Given that this reviewer does not work in the immediate research area of the paper, I found the paper very hard to follow and read. The major issues contributing to this are:

   1. Since the beginning lines, the paper takes the knowledge about efficient PI as assumed knowledge. For instance, the paper never discusses why the ReLU count is important for efficient PI. As the paper builds upon the importance of lowering the ReLU counts, I firmly believe that it should provide a reasonable amount of discussion around why the number of ReLUs are critical for efficient PI.

   2. Moreover, given the length of the paper, the structure of the writing is hard to follow. Unfortunately, instead of motivating each section properly, the paper just uses bold paragraph headings to alleviate this issue, which in my opinion does not adequately satisfy the importance of a proper motivation. Take the introduction for example. I believe that instead of having bold headings for each paragraph, it would be much easier to ensure that each paragraph follows the previous one and reads coherently. Unfortunately, this is not the case for this submission.

  3. On a minor note, I strongly recommend to the authors to use the natbib package correctly for their citations. Citations that are not part of the sentence should always be used with a parenthesis using \citep. For example, instead of writing: "_CryptoNAS Ghodsi et al. (2020) and Sphynx Cho et al. (2022a) employ neural architecture search to design ReLU-efficient baseline networks and disregard FLOP implications._" the sentence should be written as "_CryptoNAS [Ghodsi et al., 2020] and Sphynx [Cho et al., 2022a] employ neural architecture search to design ReLU-efficient baseline networks and disregard FLOP implications._" This can greatly enhance the readability of the paper.

---

> ### Author Response · Authors · 2024-02-05
> **Rebuttal to the Reviwer's comment**
>
> We thank the reviewer for their constructive feedback, which improved the quality and readability of the paper. In response, we have incorporated a new discussion at the beginning of the introduction (marked in red) to emphasize the importance of ReLU count in private inference. Additionally, we have rewritten the introduction (marked in blue) to improve the overall coherence. This includes adding a paragraph at the end that outlines the paper's organization, making it more accessible and easier for readers to follow. We also revised Section 3 to provide the context of our experimental findings and their significance for improving the efficiency of private inference.
>
> We are also thankful to the reviewer for recommending the use of the \citep format for references. We made these changes in the revised version of our manuscript.

---

### Author Response · Authors · 2024-02-05
**Summary of the Reviews**

We thank the reviewers for their constructive, insightful, and detailed feedback. We are encouraged that they appreciate our paper for the following reasons:

1. A comprehensive experimental study on SOTA PI methods, pinpointing the existing fallacies.

2. A useful benchmarking for researchers in the field.

3. Novelty of the ReLU equalization.

4. Straightforward implementation of the proposed method.

5. The relevance and strong motivation for proposed methods and optimization strategies.

6. The paper's richness in experimental content with the insightful observations and findings.

We believe that we have addressed all concerns raised by each reviewer and would be glad to clarify any outstanding issues. ```In the revised draft, rewritten text is highlighted in blue, while newly added content is marked in red.```

---

### Decision · Action_Editor_KVko · 2024-04-08

**Recommendation:** Accept as is

**Comment:**

### Summary of the paper

This paper introduces DeepReShape, a method designed to optimize neural networks for more efficient private inference (PI) by addressing ReLUs and FLOPs, which are crucial for performance. Contrary to previous work that primarily focused on minimizing ReLUs due to their perceived impact on PI latency, this study reveals that FLOPs contribute significantly to latency. They propose an approach called DeepReShape, which suggests a strategic distribution of network channels to optimize the positioning of ReLUs based on their criticality, achieving improvements in ReLU and FLOP efficiency. The networks created using DeepReShape, or HybReNets, have enhanced performance in standard PI benchmarks. Specifically, on the CIFAR-100 dataset, HybReNets achieved a 2.1% accuracy increase and a 5.2 times faster runtime at comparable ReLU counts. On the TinyImageNet dataset, they demonstrated an 8.7 times faster runtime at similar accuracy levels. This research also examines the role of network selection in prior ReLU optimization efforts and highlights the critical attributes of networks for superior PI performance.

### Summary of the discussions
This paper was initially very poorly written, as also noted by reviewers CFqf and T1uq. However, the authors did a very good job during the rebuttal period and managed to improve the writing of the paper significantly. As it stands, I think this paper reads well. The authors also, in my opinion, managed to address the other concerns related to experimental methodologies and comparisons very well, and they made a significant update to the paper. However, the Reviewer T1uq is still concerned about the CoPriv method at NeurIPS’23, which uses OT for nonlinear operations and has different (network) optimization goals. CoPriv's proposed method is designed to align with studies predominantly using Garbled Circuits; therefore, it did not adopt the Cheetah framework," which is still confusing and concerning. Although GC and OT are different techniques, they both work for private inference (the goal of the related works is the same in protecting the user's privacy). Thus, it would be useful to compare both methods. I would recommend the authors to include and add some comparisons to the CoPriv method if they can.

**Audience:**

This paper would be of interest to researchers interested in privacy and deep learning methods.  I think it is an important, very relevant and timely topic.

**Claims And Evidence:**

This paper provides extensive experiments, and it is filled with several experiments and interesting observations based on these experiments. The paper mainly focuses on computer vision tasks, but I think the findings may translate into improvements in other modalities as well. The topic is important, and the solution and approach proposed in this paper are interesting. Thus I recommend for acceptance of this paper.